# SCHENGEN receptor module drives localized ROS production and lignification in plant roots

Satoshi Fujita[1],[*],[†] (ID), Damien De Bellis[1],[2] (ID), Kai H Edel[3], Philipp Köster[3],[‡], Tonni Grube Andersen[1],[§], Emanuel Schmid-Siegert[4], Valérie Dénervaud Tendon[1], Alexandre Pfister[1], Peter Marhavý[1],[¶] (ID), Robertas Ursache[1], Verónica G Doblas[1],[††] (ID), Marie Barberon[1],[‡‡], Jean Daraspe[2], Audrey Creff[5], Gwyneth Ingram[5], Jörg Kudla[3] & Niko Geldner[1],[**] (ID)

## Abstract

Production of reactive oxygen species (ROS) by NADPH oxidases (NOXs) impacts many processes in animals and plants, and many plant receptor pathways involve rapid, NOX-dependent increases of ROS. Yet, their general reactivity has made it challenging to pinpoint the precise role and immediate molecular action of ROS. A well-understood ROS action in plants is to provide the co-substrate for lignin peroxidases in the cell wall. Lignin can be deposited with exquisite spatial control, but the underlying mechanisms have remained elusive. Here, we establish a kinase signaling relay that exerts direct, spatial control over ROS production and lignification within the cell wall. We show that polar localization of a single kinase component is crucial for pathway function. Our data indicate that an intersection of more broadly localized components allows for micrometer-scale precision of lignification and that this system is triggered through initiation of ROS production as a critical peroxidase co-substrate.

**Keywords** extracellular diffusion barriers; Casparian strips; lignin; localized ROS production; polarized signaling

**Subject Categories** Cell Adhesion, Polarity & Cytoskeleton; Development; Plant Biology

The EMBO Journal (2020) 39: e103894

## Introduction

As in animals, NADPH oxidase-produced ROS in plants is important for a multitude of processes and the number of NADPH oxidase genes (10 in *Arabidopsis*, called *RESPIRATORY BURST OXIDASE HOMOLOGs, RBOHs, A-J*) suggests a high complexity of regulation of ROS production in plants. Among its many roles, ROS-dependent regulation of plant cell wall structure and function is considered to be among its most important (Kärkönen & Kuchitsu, 2015). The cell wall is the nano-structured, sugar-based, pressure-resisting extracellular matrix of plants, and NOXs are thought to be the predominant ROS source in this compartment (also termed apoplast; Kärkönen & Kuchitsu, 2015).

A staggering number of kinases have been shown to regulate plant NOXs and the activation mechanism of NOX-dependent ROS production is well established, especially in response to microbial pattern recognition by immune receptors (Zipfel, 2014). However, the specific role and direct molecular targets of ROS during microbial pattern recognition have remained elusive (Qi *et al*, 2017). The same applies to the central role of ROS in tip growing cells, such as root hairs or pollen tubes, where ROS is thought to be part of an intricate oscillation of cell wall stiffening and loosening, aimed at allowing cell wall expansion without catastrophic collapse (Monshausen *et al*, 2007; Boisson-Dernier *et al*, 2013). In this case, ROS is proposed to be important for counteracting cell wall loosening pH decreases, but it is again unclear what direct targets of ROS would mediate cell wall stiffening. Cell wall lignification by apoplastic peroxidases can therefore be considered as the most well-established role of ROS, where the peroxidases themselves are the direct "ROS targets", using it as a co-substrate for the oxidation

1 Department of Plant Molecular Biology, Biophore, University of Lausanne, Lausanne, Switzerland
2 Electron Microscopy Facility, University of Lausanne, Lausanne, Switzerland
3 Institut für Biologie und Biotechnologie der Pflanzen, Westfälische Wilhelms-Universität Münster, Münster, Germany
4 Vital-IT Competence Center, Swiss Institute of Bioinformatics, Lausanne, Switzerland
5 Laboratoire Reproduction et Développement des Plantes, Université de Lyon, ENS de Lyon, CNRS, INRAE, Lyon, France
 *Corresponding author. Tel: +81 55 981 6799; E-mail: satoshi.fujita@nig.ac.jp
 **Corresponding author. Tel: +41 21 692 4192; E-mail: niko.geldner@unil.ch
 †Present address: National Institute of Genetics, Mishima, Shizuoka, Japan
 ‡Present address: Department of Plant and Microbial Biology, University of Zurich, Zurich, Switzerland
 §Present address: Max Planck Institute for Plant Breeding Research, Cologne, Germany
 ¶Present address: Umeå Plant Science Centre (UPSC), Department of Forest Genetics and Plant Physiology, Swedish University of Agricultural Sciences (SLU), Umeå, Sweden
 ††Present address: Institut Jean-Pierre Bourgin, INRAE, AgroParisTech, Université Paris-Saclay, Versailles, France
 ‡‡Present address: Department of Botany and Plant Biology, Quai Ernest-Ansermet, Geneva, Switzerland

of mono-lignols (Liu, 2012; Barbosa *et al*, 2019). In the case of lignification, however, a molecularly defined signaling pathway that induces ROS production during lignification has not been defined. A few years ago, our group identified a specific NADPH oxidase, RBOHF, to be required for the localized formation of lignin in the root endodermis (Lee *et al*, 2013). Lignin is a polyphenolic polymer that is generated by the radical coupling of mono-lignols, oxidized through the action of ROS-dependent peroxidases, as well as laccases (Liu, 2012). The hydrophobic lignin polymer impregnates the cellulosic cell wall of plants, rendering it unextendible and highly resistant to degradation. Lignin in the root endodermis is deposited in a central, longitudinal band around every endodermal cell. Named Casparian strips (CS), these ring-like lignin structures fuse into a supracellular network, establishing a tissue-wide, extracellular diffusion barrier (Fig 1A), analogous to epithelial tight junctions in animals (Geldner, 2013; Barberon & Geldner, 2014). Functionality of this barrier can be easily visualized by a block of penetration of a fluorescent cell wall dye, propidium iodide, into the vasculature (Alassimone *et al*, 2010; Naseer *et al*, 2012). CS localization occurs through the action of CASPARIAN STRIP DOMAIN PROTEINS (CASPs), 4-TM proteins, which form a highly scaffolded transmembrane protein platform, assembling RBOHF and cell wall peroxidases and other proteins at the Casparian strip domain (CSD; Roppolo *et al*, 2011; Hosmani *et al*, 2013; Lee *et al*, 2013). CSD formation and lignin deposition are coordinated such that the aligned rings of endodermal neighbors' fuse, leading to a supracellular network that seals the extracellular space between endodermal cells, generating a tissue-wide diffusion barrier.

More recently, we identified a pair of peptide ligands, a leucine-rich repeat receptor-like kinase (LRR-RLK) and a cytoplasmic kinase, whose phenotypes, genetic interaction, and specific subcellular localizations led us to propose that they combine into a barrier surveillance pathway. Previous reports had shown that the CIF1/2 (CASPARIAN STRIP INTEGRITY FACTORs 1/2) peptides are SCHENGEN3 (SGN3) (also called GASSHO1 (GSO1)) ligands and that the SGN1 and SGN3 kinases govern Casparian strip integrity (Pfister *et al*, 2014; Alassimone *et al*, 2016; Doblas *et al*, 2017; Nakayama *et al*, 2017). *cif1 cif2* and *sgn1* and *sgn3* mutants have similar, discontinuous CS, caused by a discontinuous CSD, as well as a conspicuous absence, or strong attenuation, of compensatory lignification and suberization observed in other CS mutants (Hosmani *et al*, 2013; Pfister *et al*, 2014; Kamiya *et al*, 2015; Doblas *et al*, 2017; Kalmbach *et al*, 2017; Li *et al*, 2017). Their phenotypic similarities suggested that these factors act in one pathway. CIF1 and 2 peptides do not express in the endodermis, where the CS is formed, but in the stele. In contrast, both SGN1 and SGN3 present specific localization on the endodermal plasma membrane; SGN3 receptor-like kinase resides along both sides of the CS, while palmitoylated SGN1 polarly localizes on cortex-facing (outer) plasma membranes (Pfister *et al*, 2014; Alassimone *et al*, 2016). Remarkably, their localization overlaps only in a small region next to the cortex-facing side of the CS (Alassimone *et al*, 2016). This would require peptides from the stele to diffuse across the CS in order to access the signaling complex. This is only possible while the CS is still permeable (Doblas *et al*, 2017; Fig 1E).

This pathway would therefore provide a mechanism that allows to probe the tissue-wide integrity of an extracellular diffusion barrier

and to respond to barrier defects through compensatory overlignification (Doblas *et al*, 2017). Here, we demonstrate that the receptor, cytoplasmic kinase, and NADPH oxidase molecularly connect into one pathway, with great resemblance to plant immune signaling pathways, but whose direct action is to locally produce ROS for spatially restricted lignification. We establish the crucial importance of the restricted subcellular localization of its components and demonstrate that stimulation of this signaling pathway additionally leads to strong transcriptional activation of target genes that further drive and sustain endodermal lignification, as well as suberization and endodermal sub-domain formation and differentiation. We thus provide a molecular circuitry in which an endogenous peptide from the stele stimulates localized signaling kinases and NADPH oxidases in the endodermis, causing extracellular ROS production at micrometer-scale precision and a precisely localized lignification of the plant cell wall.

# Results

We generated a new *cif1-2 cif2-2* double mutant allele by CRISPR-Cas9 in a pure Col background, because the previous *cif1-1 cif2-1* double mutant allele (Nakayama *et al*, 2017) was a mixture of Ws and Col alleles. The new CRISPR allele was complemented by CIF1 or CIF2 application, visualized by PI (propidium iodide) uptake assays or reconstitution of CASP1 membrane domain connectivity (Fig EV1D and E, and Appendix Fig S1A).

### Apolar SGN1 kinase leads to constitutive barrier defect signaling

Central to the barrier surveillance model is the polar localization of SGN1, which is thought to limit the potential for signal activation to the cortex side of the endodermis, requiring passage of CIF peptides across the CS region (Fig 1A and E). Consistently, application of peptide ligand to the media, leading to stimulation from the outside, causes overlignification at the cortex-facing endodermal edges (Fig 1B). In an attempt to falsify the model we had proposed and to interrogate the importance of SGN1 polar localization, we generated a SGN1 variant that localized in an apolar fashion, by adding a myristoylation (myr) and palmitoylation (palm) motifs on the N-terminus (Vermeer *et al*, 2004). This myrpalm SGN1-mCitrine (Citrine) variant was expressed under the control of the endodermis-specific CASP1 promoter, which is strongly active during Casparian strip formation and complemented the *sgn1* barrier phenotype (Fig EV1A). *In planta*, the wild-type SGN1-Citrine variant resides polarly on the cortex-facing side of endodermal cells, while myrpalm SGN1-Citrine was observed at both sides of the endodermal plasma membranes, even though preferentially accumulation at the cortex side could still be observed (Fig 1C and D). Both variants were excluded from the central position where the Casparian strip domain is formed (Fig EV1B), and the localization patterns of the two variants did not change when introgressed into *sgn1*, *sgn3*, or *cif1 cif2* mutants (Fig 1C and D). An apolar SGN1 localization would allow SGN3 to encounter SGN1 also on the stele-facing side, not only on the cortex side (Fig 1E). This should lead to constitutive signal activation in the absence of barrier defects, because the CIF peptides would now be able to access a SGN3/SGN1 signaling module on the stele-facing side without crossing the

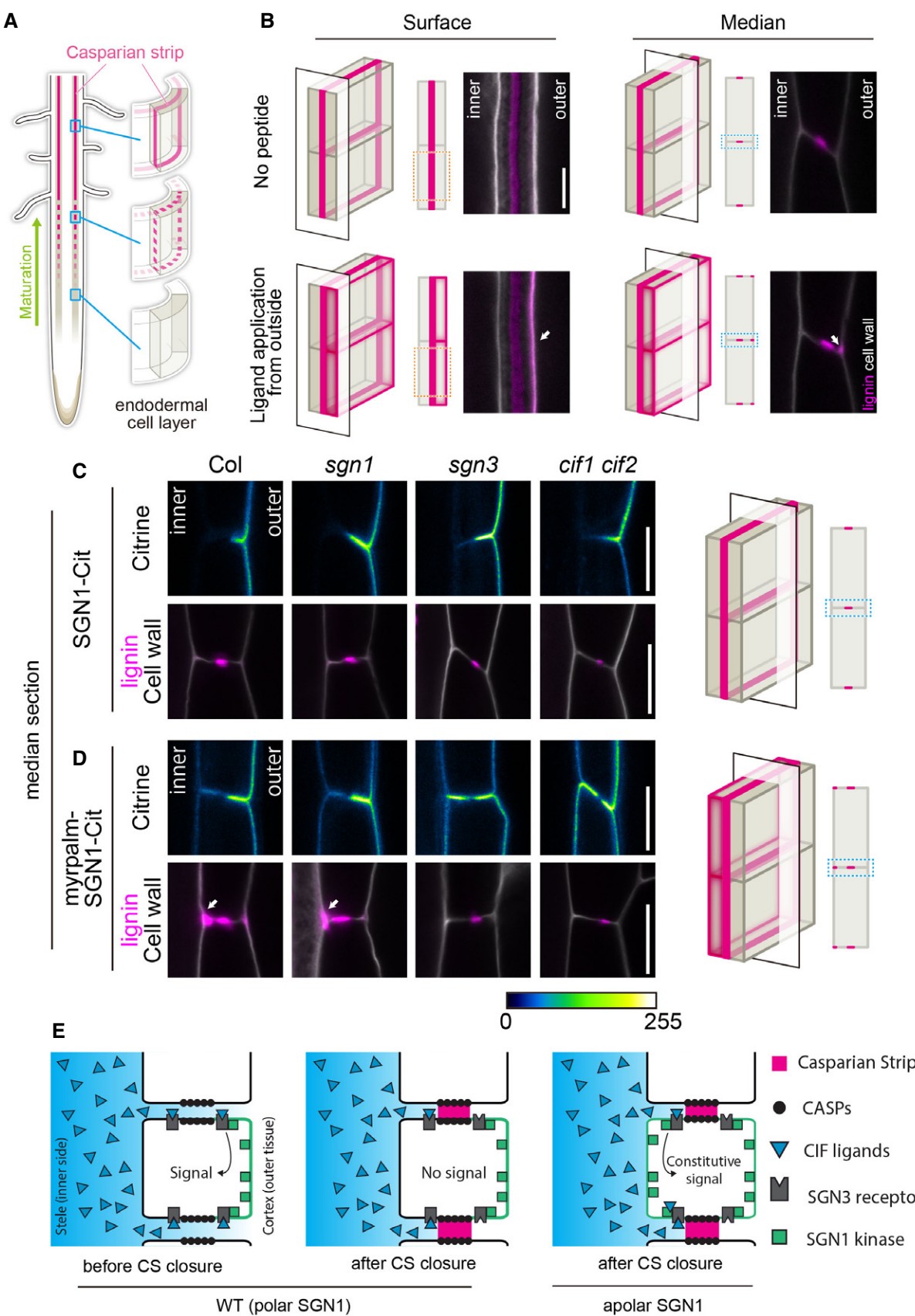

**Figure 1.**

◀

**Figure 1. Apolar SGN1 leads to ectopic lignin accumulation in endodermal cells.**

A   Schematic of Casparian strip development (magenta). Casparian strips start to appear as centrally aligned discontinuous dots in the endodermal cell layer, progressing into a network of fused rings functioning as a root apoplastic barrier.

B   Lignin accumulation patterns at endodermal surface or median positions with or without the 100 nM CIF2 (Casparian strip integrity factor 2) ligand treatment. Lignin and cellulosic (unmodified) cell walls are stained with Basic Fuchsin and Calcofluor White, shown in magenta and white, respectively. Schematics are indicating the position of optical sections in a 3D illustration. For each condition, at least 10 roots were tested and showed similar results in two independent experiments. White arrows indicate sites of excess lignification on the cortex-facing (outer) side. Scale Bar = 5 μm.

C, D   Localization of SGN1-Citrine and lignin deposition patterns in pCASP1::SGN1-Citrine lines in wild-type (Col) and different mutant backgrounds (sgn1, sgn3, cif1 cif2) (C). myrpalm-SGN1-Citrine localization and lignin deposition patterns in pCASP1::myrpalm-SGN1-Citrine lines (D). Lignin (Basic Fuchsin) and cell walls (Calcofluor White) are shown in magenta and white, respectively. For this experiment, two or three independent lines were tested. From each transgenic line, 2 positions from 12 roots were observed and representative pictures are shown in the figure. Schematics are indicating the position of optical sections in a 3D illustration. White arrows in (D) highlight excess lignification on the pericycle-facing (inner) side. Scale bars = 5 μm.

E   Schematic illustrating how signal activation can be governed by SGN1 localization and peptide ligand diffusion from the stele.

Data information: "inner" designates the stele-facing endodermal surface, "outer", the cortex-facing surface.

barrier. Indeed, we found that the apolar SGN1 variant caused both, ectopic lignin deposition and precocious suberization in endodermal cells (Figs 1D and EV1F and G), as previously described for endodermal barrier mutants. Yet, no barrier defect was observed in the lines complemented with apolar SGN1 (Fig EV1A) and, consistently, we found CASP1-mCherry distribution to be normal in these lines, forming a continuous band in the central position of the endodermal cells, indistinguishable from wild type (Fig EV1B). This indicates that the presence of SGN1 at the plasma membrane to the inside of the CS leads to signaling in the absence of barrier defects. CASP1 promoter-driven wild-type SGN1 lines as a control complemented the mutant (Fig EV1A) and did not cause any changes in lignin accumulation pattern (Fig 1C). Interestingly, the apolar SGN1 lines accumulated lignin mainly on the stele-facing edges of the endodermal cell walls, as opposed to the cortical lignin deposition observed by ectopic ligand treatment (compare Fig 1B with 1D, arrows). The ectopic lignin deposition in apolar SGN1 lines was fully dependent on the presence of receptor and ligand; as in both sgn3 and cif1 cif2 mutants, no excess lignification at the stele-facing side could be observed in apolar SGN1 lines (Fig 1D). This strongly suggests that mislocalized SGN1 does not become constitutively active, but leads to continuous, ectopic transduction of CIF1/2 signals through SGN3. An apolar, but kinase-dead variant of SGN1 was also unable to induce ectopic lignification (Fig EV1C), suggesting that a phosphorylation relay downstream of SGN1 is necessary for lignification.

## SGN1 is a downstream component of the CIF/SGN3 pathway

Previous data showed that sgn1 is less sensitive to high doses of externally applied CIF peptide, consistent with a role of SGN1 downstream of the SGN3 receptor (Doblas et al, 2017), at least with respect to CIF-induced excess lignification. In order to address whether SGN1 is indeed a generally required downstream component of the CIF/SGN3 pathway during CS formation, we evaluated whether the sgn1 mutant is resistant to complementation by CIF2 peptide treatment. In contrast to the full complementation of the discontinuous CS domain of cif1 cif2 double mutants, the sgn1 mutant did not show full restoration of domain integrity, neither on 10 nM nor on 100 nM CIF2 medium (Fig 2A, Appendix Fig S1A). While some degree of rescue occurred, only about 50% of discontinuities were rescued. This was corroborated by testing CS functionality using the propidium iodide (PI) assay. Only a weak complementation of barrier formation was observed

even when grown on 100 nM CIF2 medium (Fig 2B). Our results indicate that SGN1 functions downstream of CIFs/SGN3, but suggest that additional factors can partially compensate for its absence, most probably homologs of the extended RLCKVII family to which SGN1 belongs.

We then tested for direct connectivity between SGN3 receptor kinase and SGN1 by carrying out an in vitro kinase assay. Some RLCKVII members are reported to be phosphorylated and activated by LRR receptor kinases (Lu et al, 2010; Kim et al, 2011). A glutathione S-transferase (GST)-fused SGN3 kinase domain was incubated with a kinase-dead form of a trigger factor (TF)-SGN1 fusion protein—which does not have autophosphorylation activity—in the presence of radioactive ATP. Kinase-dead TF-SGN1 was efficiently phosphorylated by SGN3 kinase domain, but not by a kinase-dead form of SGN3 in vitro (Fig 2C, Appendix Fig S1B). This further supports our model that SGN1 is a direct downstream component of the SGN3/CIF pathway.

## Two NADPH oxidases are absolutely required for CIF-induced lignification in the endodermis

Our generation of an apolar SGN1 thus appears to have reconstituted a functional CIF/SGN3 pathway at the stele-facing (inner) endodermal surface, causing ectopic lignification in the absence of barrier defects. Yet, the second intriguing aspect of this manipulation is the observation that lignification occurs almost exclusively at the inner endodermal edges (Fig 1D), the side where endogenous CIF peptide must be assumed to be present. External treatment, by contrast, leads to predominant lignification at the outer endodermal edges (Fig 1B). This surprising spatial correlation between the site of signal perception and localized lignification suggests a very direct molecular connection between the two events that would allow to maintain spatial information. In a cell primed for lignification, i.e., with mono-lignol substrates available and polymerizing enzymes expressed, lignification could be simply "switched on" by activating ROS production through NADPH oxidases. Previously, we had found one of the NOXs, RBOHF, to be crucial for lignification at the CS. Intriguingly, RBOHF is the only transmembrane protein known to accumulate at the Casparian strip domain, safe the CASPs themselves (Lee et al, 2013; Fig 3A). Moreover, homologous NADPH oxidases, such as RBOHB or RBOHD, also present in the endodermal plasma membrane, are excluded from this domain (Lee et al, 2013; Fig 3A). We therefore asked whether CIF peptides would induce lignification through activation of RBOHF as a

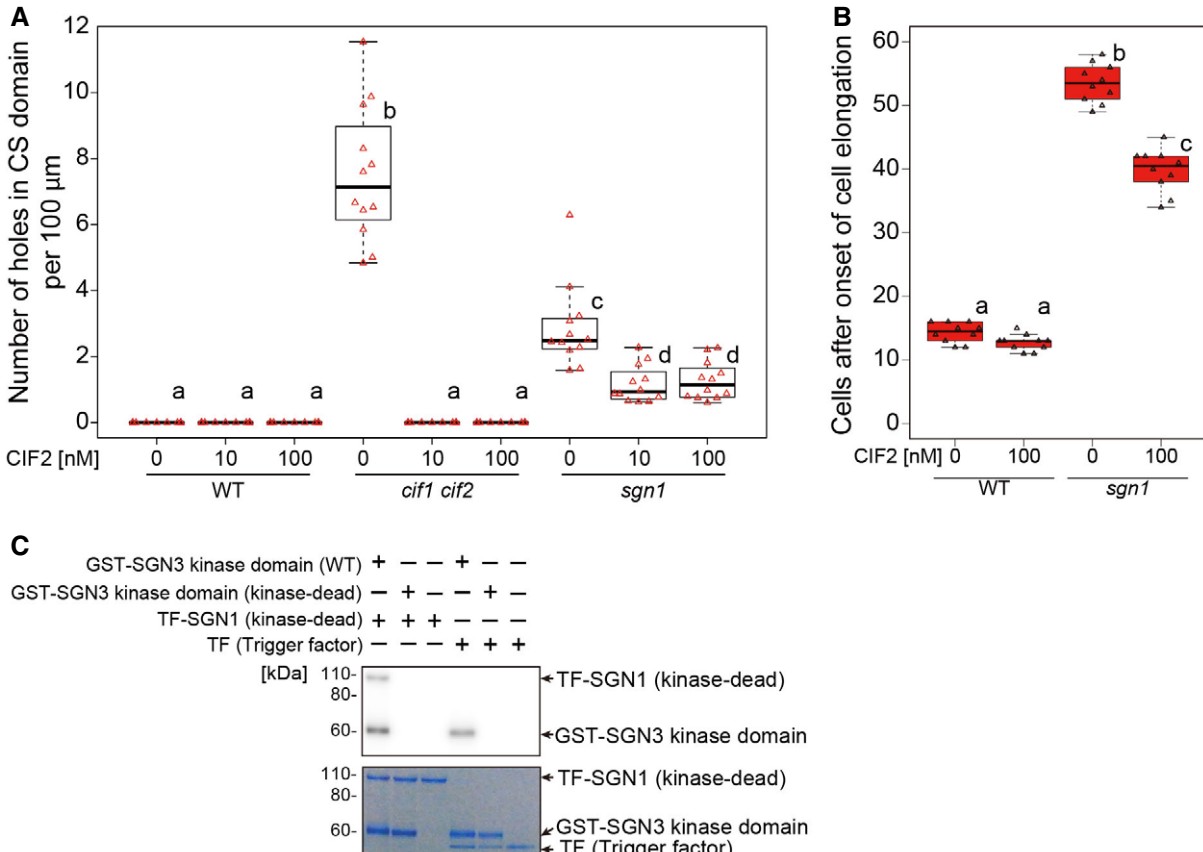

**Figure 2. SGN1 acts as a transducer of CIF2 signaling and is phosphorylated by the SGN3 receptor.**

A Quantification of defects in CSD formation as number of holes per 100 μm in the CASP1-GFP domain at around 10 cells after onset of CASP1-GFP expression in 5-day-old seedlings. In the box plot, boxes indicate ranges from first to third quartiles, and the bold central lines display median. Upper and lower whiskers extend to maximum or minimum values no further than 1.5 times IQR (interquartile range, the distance between the first and third quartiles). One-way ANOVA was performed followed by Tukey's test. Different letters show significant statistical differences ($P < 0.05$, one-way ANOVA and Tukey's test, 12 roots in total were observed for each condition in two independent assays).

B Propidium iodide (PI) penetration assay in the presence or the absence of CIF2. CS barrier function was scored as exclusion of PI signal from the inner side of endodermal cells. In the box plot, boxes indicate ranges from first to third quartiles, and the bold central lines display median. Upper and lower whiskers extend to the maximum or minimum values no further than 1.5 times IQR. Different letters show significant statistical differences ($P < 0.05$, one-way ANOVA and Tukey's test. During two independent experiments, 10 roots in total were tested for each condition).

C [γ-$^{32}$P]ATP radioactive *in vitro* kinase assay of SGN3 kinase domain against SGN1. Autoradiograph is shown on top. Coomassie-stained gel below illustrates presence and equal loading of recombinant proteins. Note that a kinase-dead SGN1 variant was used to avoid autophosphorylation activity of SGN1. Also note that trigger factor represents a very big tag protein, accounting for the high migration of TF-SGN1. Representative result of three independent experiments is shown.

downstream component. To our surprise, CIF treatment still led to induced lignification in *rbohF*, despite the fact that RBOHF is strictly required for CS lignification in untreated conditions. Yet, when RBOHD was knocked-out in addition to RBOHF, a complete absence of lignification upon CIF treatment was observed (Fig 3B). The fact that RBOHD single mutant showed defects neither in developmental CS lignification, nor in CIF-induced lignification, indicates that RBOHF is required for both processes, but that upon strong stimulation with exogenous CIF peptide application, RBOHD can be additionally used. Indeed, using the general NADPH oxidase inhibitor diphenyleneiodonium (DPI) in short-term co-treatment with CIF was also able to fully block CIF-induced ectopic lignification in wild type with intact Casparian strips, further supporting that NADPH oxidases act downstream in the CIF/SGN3 pathway (Fig EV2A).

**CIF2 induces highly localized ROS production through RBOHF and RBOHD**

The direct activity of NOX enzymes is not lignification, but production of superoxide ($O_2^-$) that becomes dismutated to hydrogen peroxide ($H_2O_2$) in the apoplast (Kärkönen & Kuchitsu, 2015). We had previously established that the different subcellular distribution of the SGN3 receptor and the SGN1 kinase only intersects at a very restricted domain at the outer (cortex-facing) edge of the CS (Alassimone *et al*, 2016; Fig 4A). We therefore attempted to visualize whether ROS might be produced locally in response to CIF treatment. Many ways exist to visualize and quantify ROS, but only few allow for high spatial resolution and for discrimination between extracellular and intracellular ROS. An older method, based on ROS-induced cerium precipitation that can be detected

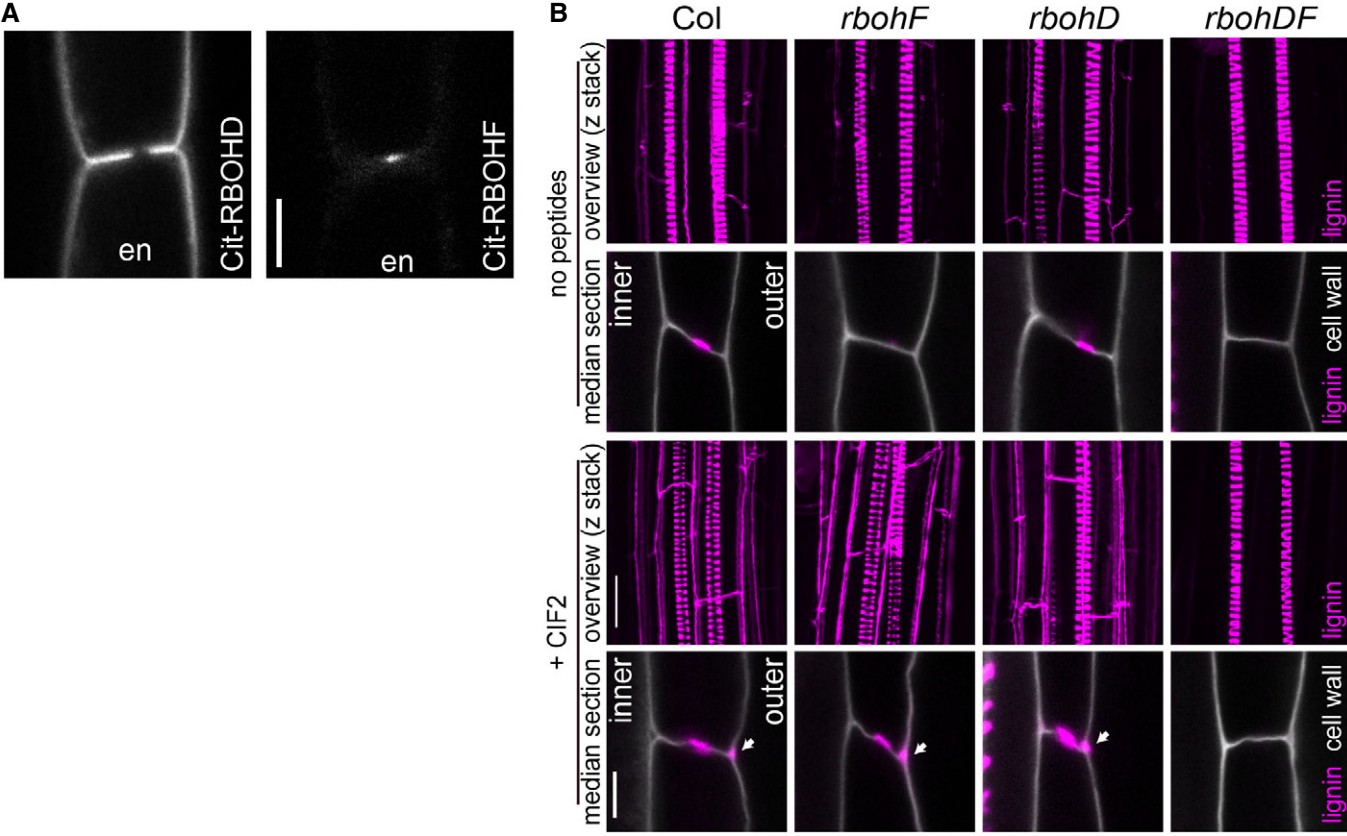

**Figure 3. Both RBOHD and F are required for CIF2-induced excess lignin accumulation.**

A Localization of Citrine-RBOHD (left) or Citrine-RBOHF (right) in endodermal (en) cells. Both proteins were expressed under the control of pCASP1, an endodermis-specific promoter. Representative pictures are shown; 2 positions from 10 roots for each transgenic line were inspected. Scale bar = 5 μm.

B Lignin accumulation in WT and *rbohD and rbohF* single mutants and a double mutant with or without 2-h 100 nM CIF2 peptide treatment. Arrowheads indicate excess lignification. Pictures are shown as overviews (maximum projection) or median sections. Lignin and cell walls are shown with magenta (stained with Basic Fuchsin) and gray (stained with Calcofluor White), respectively. Representative pictures are shown; 12 roots (overview) and 2 positions in 12 roots (median section) were inspected. Scale bars = 20 μm (lignin overviews), 5 μm (median sections). "inner" designates the stele-facing endodermal surface, "outer", the cortex-facing surface.

using transmission electron microcopy (TEM), has been used by us previously to demonstrate that a highly localized ROS production indeed occurs at the CS and is dependent on NOX activity (Bestwick *et al*, 1997; Lee *et al*, 2013; Fig 4B). Using this method, we could detect a strong ROS production in response to CIF2, exclusively at the regions of endodermal–endodermal cell walls outside of the CS, but nowhere along the endodermis-cortex cell walls, which are equally reached by the cerium chloride and where NOX enzymes are also present in the plasma membrane (Fig 4B and C). Based on this striking spatial coincidence between the SGN3/SGN1 overlap region (Fig 4A; Alassimone *et al*, 2016) and CIF2-induced ROS production, we developed a procedure to quantitatively assess cerium precipitates and checked whether this localized ROS production is indeed dependent on the SGN3 pathway (Figs 4D and E, and EV3A, see also Experimental Procedures section). For SGN3, we found that already steady-state ROS production was undetectable in the mutant and that there was no increase upon CIF2 treatment (Fig 4D and E). The *sgn1* mutant showed lower, but still detectable steady-state ROS levels, but no significant increase upon CIF2 treatment (Fig 4D and E). Thus, the

highly localized ROS accumulation induced by CIF2 is entirely dependent on the localized SGN3/SGN1 receptor module, suggesting that the localization of the module determines the spatial extent of ROS production. As in the case of lignification, we observed that CIF2-induced ROS could be produced by either RBOHF or RBOHD, as only the double mutant caused a complete absence of ROS after CIF stimulation. As expected, but not previously demonstrated, the steady-state ROS production at the CS observed before stimulation was exclusively dependent on RBOHF, but not RBOHD (Fig 4D and E).

In contrast to lignin polymerization, suberin might not require NADPH oxidase activity. It was shown previously that CIF2 triggers excess suberization in WT (Fig EV2B and C) (Doblas *et al*, 2017). We found excess suberin deposition in *rbohD* and *rbohF* upon peptide treatment, although a slight enhancement is already observed in *rbohF*. In *rbohDF*, excess suberin deposition is very strong, even without treatment, likely due to a strong activation of the surveillance system due to complete absence of a CS (Fig EV2B and C). These data indicate that CIF2-triggered suberin accumulation is not affected by NADPH oxidases and suggests existence of a

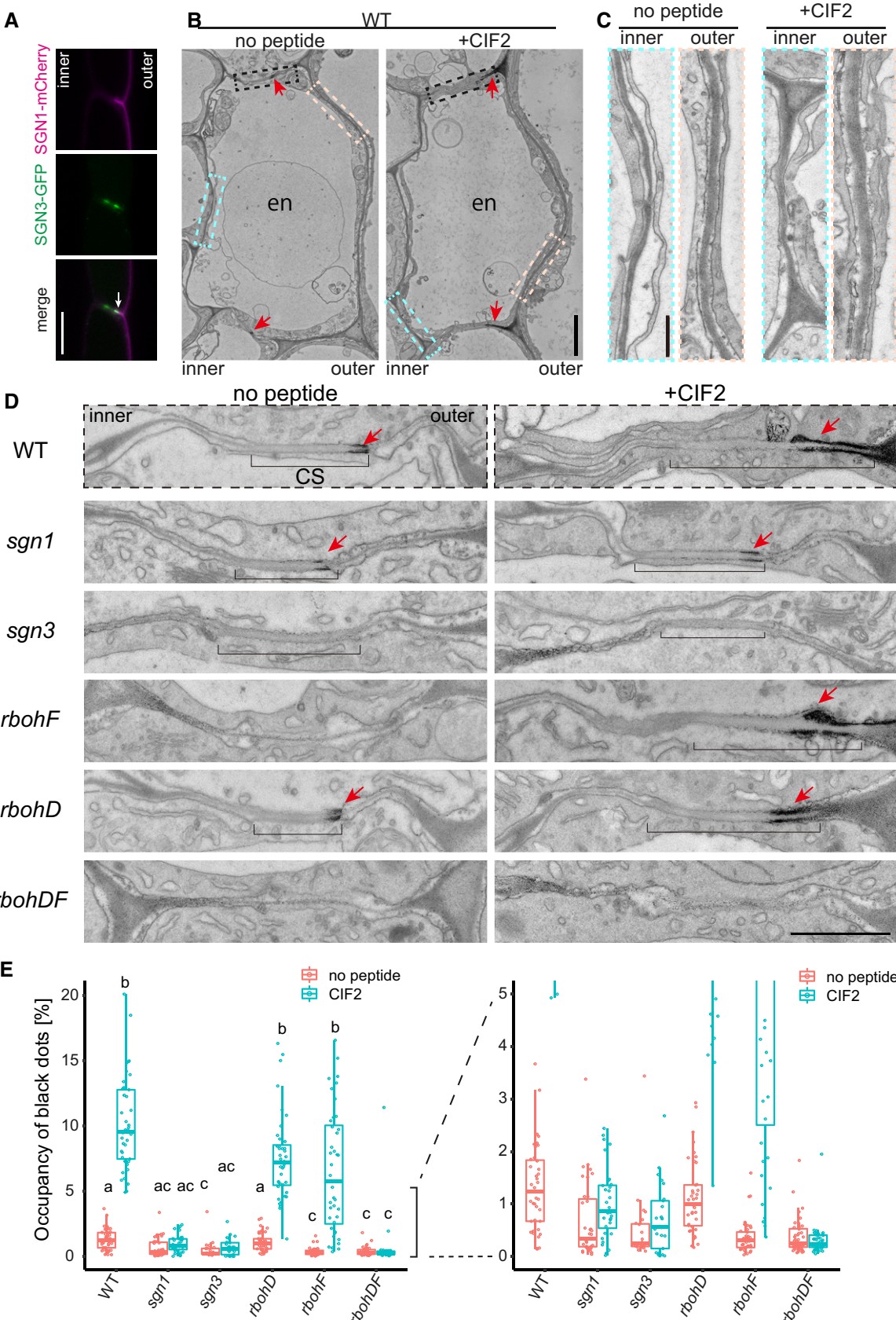

**Figure 4.**

Figure 4. ROS production is enhanced by SGN3/CIFs and requires RBOHD and F.

A Co-visualization of *pCASP1::SGN1-mCherry* and *pCASP1::SGN3-GFP*. Note that their respective localization at the PM has only restricted overlap at the cortical side of the CS domain (Arrow). Representative picture is shown; 3–6 roots in four different transgenic lines were inspected. Scale bar = 10 μm. "inner" designates the stele-facing endodermal surface, "outer", the cortex-facing surface.

B, C Overview of endodermal cells after with or without 1 μM CIF2 treatment for 24 h. Red arrowheads are indicating ROS production sites, and boxes in dotted lines are corresponding to the regions in (sky blue and creme boxes in C and black boxes in D). Similar patterns were obtained in 39 or 41 cells from 5 roots with or without the peptide treatment. Scale bars = 500 nm (B) and 100 nm (C).

D *In situ* $H_2O_2$ detection at Casparian strips in WT, *sgn3*, *sgn1*, *rbohF*, *rbohD*, and *rbohDF* with or without 24-h treatment of 1 μM CIF2. Brackets and arrowheads indicate Casparian strips (seen as uniformly whitish cell wall areas) and $H_2O_2$ production sites (black area), respectively. Scale bar = 1000 nm.

ROS-independent branch of the SCHENGEN pathway that regulates this process (see below).

## SGN1 can directly activate RBOHF and RBOHD via phosphorylation

The above data strongly suggest a direct connection between the SGN3/SGN1 kinase module and the two NOX enzymes. We therefore conducted an *in vitro* kinase assay in order to ask whether SGN1 can directly phosphorylate the N-terminal cytoplasmic region of RBOHF and RBOHD. We found that recombinantly expressed TF-SGN1 could phosphorylate both the recombinant N-terminal part of RBOHF and RBOHD (Fig 5A). Plant NOX regulation has been intensively studied and shown to be highly complex, requiring possibly interdependent activities of kinases, as well as small GTPases (Kadota *et al*, 2015). We therefore tested whether SGN1 might be sufficient for activation of RBOHF activity in a cellular context. To do so, we made use of a heterologous reconstitution system in human HEK293T cells, which show very low endogenous ROS production and for which it had been previously demonstrated that plant NADPH oxidases can be expressed and their activation mechanism be studied (Han *et al*, 2019). As a positive control, the previously described calcium-dependent kinase complex of calcineurin B-like (CBL) interacting protein kinases 26 (CIPK26) and CBL1 was used and shown to be active (Fig 5B). When expressing wild-type SGN1 in this cell line, we noticed that it did not activate RBOHF or RBOHD, but that it also did not localize to the plasma membrane as in plant cells (Fig EV4A). However, when we used the functional, constitutively plasma membrane-localized myrpalm-SGN1 version, a significant induction of ROS production was observed for RBOHF and to a lesser extent for RBOHD (Figs 5B and EV4B).

## CIF peptide-induced CASP domain growth requires new protein synthesis, but not ROS production

The direct phospho-relay from SGN3 receptor, to SGN1 kinase to RBOHF and RBOHD outlined above draws a direct molecular connection from perception of a peptide hormone stimulus to cell wall lignification. Moreover, it accounts for both, the highly localized ROS production that we observe upon CIF stimulation and the observation that localization of lignification is correlated with the site of active SGN3 signaling. Yet, the massive enhancement of lignification observed upon CIF stimulation (Fig 3B), the increase of CASP accumulation, and ectopic patch formation (Appendix Fig S1A), as well as the non-localized formation of precocious and enhanced suberin (Fig EV2B and C), are additional outcomes of

SGN3 pathway stimulation and should be driven by transcriptional changes. Indeed, some degree of transcriptional upregulation of CASP genes has been reported previously upon CIF1 treatment (Nakayama *et al*, 2017). CIF stimulation does not only lead to enhanced accumulation and ectopic patches of CASP1-GFP, absence of SGN3 signaling but also lead to discontinuous CASP1-GFP signals that could be explained by insufficient amount of CASPs and other factors being produced during endodermal differentiation (Fig 6A; Pfister *et al*, 2014; Doblas *et al*, 2017). Neither the single, nor the double NADPH oxidase mutants display discontinuous CASP1-GFP signals, indicating that ROS production is not required for this aspect of the CIF/SGN3 pathway (Fig 6A). In order to directly demonstrate that formation of a continuous CASP domain requires newly formed gene products, we treated our *cif1 cif2* double mutant with CIF2 peptide in the presence or absence of protein synthesis inhibitor cycloheximide (CHX). CASP1-GFP signal strongly increased during 8 h of CIF treatment, during which the discontinuous CASP1-GFP domains became continuous. This effect was abrogated by CHX treatment (Fig 6B and C, Movie EV1).

## The NADPH oxidase-independent branch of the CIF/SGN3 pathway is associated with MAP kinase stimulation and causes strong activation of gene expression

In pattern-triggered immune receptor signaling, gene activation is thought to depend in large parts on activation of mitogen-activated protein (MAP) kinases (Dodds & Rathjen, 2010). We therefore tested whether CIF treatment leads to MAP kinase phosphorylation and indeed found that CIFs can induce MAP kinase phosphorylation in a SGN3-dependent manner, further extending the molecular parallels between immune receptor signaling and the CIF/SGN3 pathway and suggesting that gene induction in the CIF pathway might equally depend on MAP kinase signaling (Fig EV5B).

We then undertook an RNA profiling of seedling roots at different time points (30, 120, and 480 min) after CIF stimulation, using wild-type, *cif1 cif2,* and *sgn3* mutants as genotypes. A total of 930 genes were found to be differentially expressed across any of the genotypes and time points combined, using a stringent cut-off (adj.-P val. <= 0.05; logFC >= 1 or logFC <= −1) (Table EV1). After normalization of batch effects, the three replicates clustered closely with a large degree of variance explained by the peptide treatment in wild type and *cif1 cif2* (Fig EV5A). Wild-type and *cif1 cif2* samples showed nearly identical responses (co-relation efficiency values were 0.90, 0.89, and 0.94 at 30, 120, and 480 min, respectively), with *cif1 cif2* displaying a slightly stronger overall amplitude

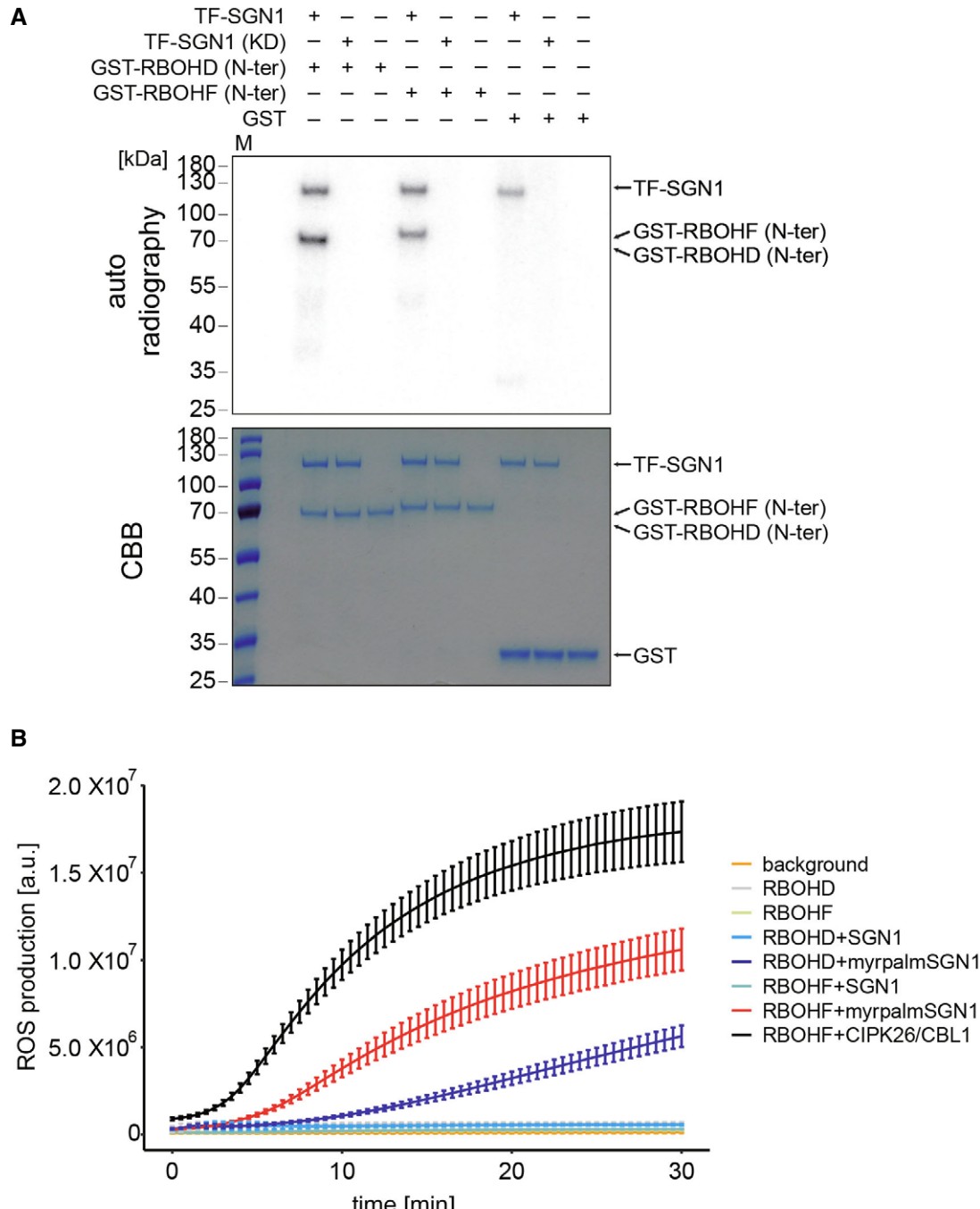

**Figure 5. SGN1 directly activates NADPH oxidases in a cellular context.**

A [γ-$^{32}$P]ATP radioactive *in vitro* kinase assay of TF-SGN1 against GST-N-terminal cytoplasmic domains of RBOHD or F. Autoradiograph is shown on top. Coomassie-stained gel below illustrates presence and equal loading of recombinant proteins. Experiments were done independently three times with similar results.

B HEK293T cell-based NOX activation assay. Cells were transfected with the indicated plasmid combinations. The phosphatase inhibitor Calyculin A was added directly before the start of the measurements. Each data point represents the mean of six wells analyzed in parallel; bars indicate SD. Experiments were repeated three times (another set of results is shown in Fig EV4B).

(Fig 6D). Importantly, *sgn3* had virtually no differentially expressed genes across treatments, indicating the specificity of the CIF responses and further corroborating that SGN3 is the single, relevant receptor for CIF responses in roots (Figs 6D and EV5A, Table EV1).

Our data extend on the previous data of Nakayama *et al* (2017), by showing that all 5 *CASP* genes are differentially regulated upon CIF treatment (Fig EV5C). Moreover, we observed upregulation of *MYB36*, a central transcription factor for Casparian strip formation

and CASP expression, thus potentially accounting for the increases in *CASP1-4* expression (Kamiya *et al*, 2015; Liberman *et al*, 2015; Fig EV5C). The fact that CIF stimulates *MYB36* expression also is consistent with the recent report that CIF treatment can enhance ectopic endodermal differentiation, driven by overexpression of the SHR transcription factor (Drapek *et al*, 2018).

Of the four response clusters defined by k-means clustering, the two largest clusters (1 and 3) group early response genes (cluster 1)

and later response genes (cluster 3). Cluster 2 groups a smaller fraction of genes showing strong and largely sustained responses over the time scale of the experiment, while cluster four groups a minor fraction of genes downregulated at later time points. GO term analysis indicates that many of the most significant, overrepresented terms in the "early response" (1) and "strong and sustained" (2) gene clusters are related to immune and defense responses (response to chitin, bacterium, callose, etc.), as well as responses to

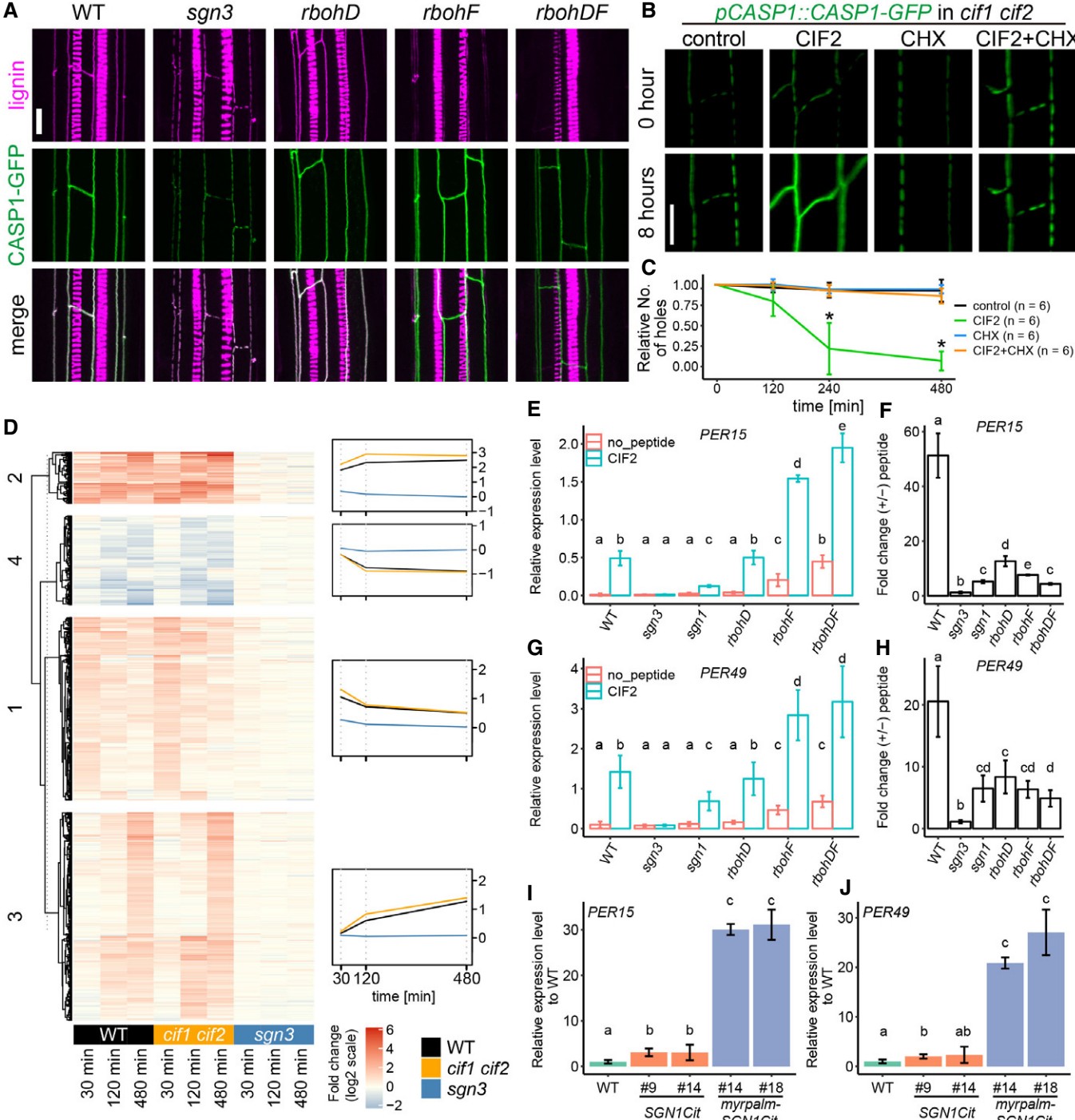

**Figure 6.**

**Figure 6.  CIF2 induces large-scale transcriptional changes for cell wall remodeling.**

A   CASP1-GFP and lignin deposition in WT, *sgn3*, *rbohD*, *rbohF*, and *rbohDF*. CASP1-GFP and lignin (fuchsin) are presented in green and magenta, respectively. Pictures were obtained from more than 10 roots from each background with similar results. Scale bar = 10 μm.

B   Time lapse imaging of single or co-treatment of 10 nM CIF2 with 25 μM cycloheximide (CHX) on CASP1-GFP in *cif1 cif2*. Seedlings were pretreated with or without CHX for 30 min and transferred onto each medium. Scale bar = 10 μm (see also Movie EV1).

C   Quantification of (B). Relative numbers of holes in CASP1-GFP domain after single or co-treatment with CIF2 or CHX from the pictures in (B). Bars are SD. * indicates statistical significance from all other conditions ($P < 0.01$) after one-way ANOVA and Tukey test. Six roots in total for each condition were observed during two independent tests.

D   Fold change of 930 genes ($P < 0.05$ and $\log_2$(fold change) $\geq 1$ or $\leq -1$ at least one time point in one genotype) after CIF2 treatment at indicated time points in WT, *cif1,2*, and *sgn3*. Degree of the fold changes is shown in color code as indicated.

E   Relative expression levels of *PER15* to *CLATHRIN* control in each genotype with or without 2-h CIF2 treatment. Bars are SD ($n = 3$). Different characters indicate statistically significant differences ($P < 0.01$, ANOVA and Tukey test).

F   Fold changes of *PER15* in each genotype with or without 2-h CIF2 treatment. Bars are SD ($n = 3$). Different characters indicate statistical significance differences ($P < 0.01$, ANOVA and Tukey test).

G   Relative expression levels of *PER49* to *CLATHRIN* control in each genotype with or without 2-h CIF2 treatment. Bars are SD ($n = 3$). Different characters indicate statistically significant differences ($P < 0.01$, ANOVA and Tukey test).

H   Fold changes of *PER49* in each genotype with or without 2-h CIF2 treatment. Bars are SD ($n = 3$). Different characters indicate statistically significant differences ($P < 0.01$, ANOVA and Tukey test).

I   Relative fold changes of *PER15* in *pCASP1::SGN1-Citrine* and *pCASP1::myrpalm-SGN1-Citrine* compared to the expression level in WT. Bars are SD ($n = 3$). Different characters indicate statistically significant differences ($P < 0.01$, ANOVA and Tukey test).

J   Relative fold changes of *PER49* in *pCASP1::SGN1-Citrine* and *pCASP1::myrpalm-SGN1-Citrine* compared to the expression level in WT. Bars are SD ($n = 3$). Different characters indicate statistically significant differences ($P < 0.01$, ANOVA and Tukey test).

oxidative stress (Table EV2). The top GO categories of the late response cluster (3) were suberin/cutin biosynthesis, supporting our observation that suberin accumulation is a late effect of CIF stimulation (Figs EV2B and C, and EV5D; Doblas *et al*, 2017). Additional, enriched terms were related to oxidative stress and cell wall remodeling, nicely fitting the CIF2-induced ROS production we observed. Intriguingly, the cluster showing late downregulation of gene expression contains many overrepresented GO terms related to a wide variety of transport processes, reaching from water, nitrate, and ammonium transport, to primary and secondary metabolite or hormone transport (Fig 6D, Table EV2). We interpret this as a consequence of CIF-induced lignification and suberization, which must profoundly impact endodermal ability for shuttling organic and inorganic compounds, as well as signaling molecules between stele and cortical root cell layers (Barberon & Geldner, 2014).

While GO terms related to lignification were also overrepresented, they were not as highly ranked as expected from the strong induction of lignification upon CIF treatment. A number of reasons might account for this: Firstly, GO terms for pathogen, defense and oxidative stress responses, are all bound to overlap with and contain genes that mediate lignification. Moreover, already elevated levels of developmental, lignin-related gene expression in the endodermis might lead to less-pronounced fold changes. Finally, the multiple roles for lignin biosynthetic genes in secondary metabolism further obscure clear categorization. Nevertheless, upregulation of lignin biosynthesis by CIF treatment became evident when clustering a set of laccases and peroxidases, the two main families of enzymes implicated in lignin polymerization in the apoplast (Liu, 2012). A significant number of both are strongly upregulated (Fig EV5E and F). Moreover, *MYB15*, a transcription factor shown to be involved in stress and MAMP(microbe-associated molecular pattern)-induced lignification (Chezem *et al*, 2017), is about sixfold induced after 30 min and about 17-fold after 120 min in both treated wild type and *cif1 cif2*, nicely correlating with the upregulation of peroxidases and laccases (Fig EV5E and F, Table EV1). We then made use of two of the most highly differentially expressed genes upon CIF2 treatment

(*PER15* and *PER49*) in order to establish whether SGN1 and the NADPH oxidases are also required for gene regulation downstream of SGN3 activation. In qPCR analysis, both peroxidase genes still showed a slight, but significant upregulation upon CIF treatment in the *sgn1* mutant (compared to the complete absence of response in *sgn3*) (Fig 6E–H), clearly indicating that, as for ROS production, SGN1 is an important, yet not absolutely required, downstream component in CIF-regulated gene expression. When directly plotting peroxidase fold inductions in *rbohF*, *rbohD,* and double mutant, a seemingly strong attenuation was observed (Fig 6F and H). However, when independently plotting gene expression in treated and untreated conditions (Fig 6E and G), this lower fold induction is largely explained by an already significant basal increase in marker gene expression in *rbohF* and *rbohD rbohF* double mutants. Indeed, both mutants display an absence of Casparian strips, which should lead to an endogenous stimulation of the SGN3 pathway, nicely supporting the barrier surveillance model. Absolute expression of both marker genes in the NADPH oxidase mutants is in effect higher than in wild type after CIF treatment, leading us to conclude that ROS production and gene activation in response to CIF are two independent branches of this signaling pathway, with the branching occurring downstream of the SGN1 kinase. We finally wanted to know whether the mislocalization of the SGN1 kinase reported initially is only affecting CIF/SGN3 signaling at the plasma membrane, or whether it also affects gene activation. We found that both *PER15* (Fig 6I) and *PER49* (Fig 6J) are dramatically upregulated in the non-polar SGN1 lines, independent of exogenous peptide application, strongly corroborating our model whereby the SCHENGEN pathway function crucially depends on the correct subcellular localization of its downstream kinase.

## Discussion

The data presented here sketch out an entire signaling pathway. Previously, SGN3 had been established as the receptor for CIF1 and

2, but its connection to SGN1 had exclusively been based on genetic evidence. Here, we show that SGN3, SGN1, and RBOHD/F are, biochemically and functionally, part of a signal transduction chain, leading directly from localized peptide perception to localized ROS production and lignification (Fig 7A). Moreover, the pathway branches downstream of SGN1, leading to MAP kinase activation and stimulation of gene expression. Some of the most strongly induced genes being peroxidases and laccases would further enhance and sustain lignification. The SCHENGEN pathway therefore elegantly integrates fast, plasma membrane-based responses that maintain positional information (ROS and lignin are produced close to where the ligand is perceived) and slower, gene expression-based responses, which have lost positional information, but would allow to enhance and maintain the ROS burst-controlled activation of peroxidases (Fig 7A).

The SCHENGEN pathway bears a striking overall resemblance to well-established signaling pathways for perception of MAMPs (Fig 7B). In MAMP perception, structurally similar receptor kinases, such as FLS2 or EFR, bind to microbial patterns and interact with the SERK family of co-receptors (Chinchilla *et al*, 2007), as is the case for SGN3 (Okuda *et al*, 2020). The signal is then transduced through kinases of the RLCKVII family, such as BIK1 or PBLs, which are homologs of SGN1 (Liang & Zhou, 2018). BIK1 in turn was shown to phosphorylate RBOHD, driving the well-described MAMP-induced ROS burst (Kadota *et al*, 2014; Li *et al*, 2014). Moreover, it has recently been shown that kinases of the RLCKVII family directly phosphorylate MAPKKKs, which now mechanistically explains how MAMP perception induces MAP kinase phosphorylation (Bi *et al*, 2018). It is thus tempting to speculate that the SCHENGEN pathway represents an ancient neo-functionalization of an immune receptor pathway. Indeed, this has been proposed recently and it was pointed out that the closest receptor homologs to SGN3 and GSO2 are PEPR1 and PEPR2 (Creff *et al*, 2019). The latter are receptors to an endogenous plant peptides (AtPEPs), whose activities resemble that of MAMPs and are best thought of as "phytocytokines", i.e., agents able to induce an immune-like response in cells that have not yet encountered MAMPs (Gust *et al*, 2017). Such an original phytocytokine might then have been neo-functionalized to induce pre-formed defensive cell wall barriers in a developmental context, in the absence of any actual biotic or abiotic stress. MAMP perception and MAMP-induced ROS production are found already in mosses and clearly precede the wide-spread adoption of lignin as a major, cell wall-reinforcing polymer (Weng & Chapple, 2010; Bressendorff *et al*, 2016). Moreover, lignin was speculated to originate from phenylpropanoid-derived defense compounds, making it plausible that immune signaling pathways have been at the origin of developmental regulation of lignification. The intriguing innovation of the SCHENGEN pathway would then reside in the subcellular arrangement of its signaling components, especially that of SGN1. Indeed, we demonstrate that polar

localization of SGN1 to the outside of the Casparian strip domain is crucial for barrier function—since its simple mislocalization to the inside leads to ligand- and receptor-dependent overlignification at inner cell corners. Thus, the only feature that arrests signaling in wild type is the formation of a lignified diffusion barrier in the cell wall, preventing access of the stele-produced peptides to SGN3 receptors at the outer domain—as the only population able to stimulate the polar SGN1 kinase. Since we have shown here that a main read-out of the SCHENGEN pathway is cell wall lignification itself, this designs a fascinating, spatial negative-feedback loop, in which SCHENGEN pathway-stimulated lignification feedback regulates itself once enough lignin has been produced to form a tight diffusion barrier and to fully prevent further CIF peptide penetration to the outside.

It will be important to further describe the extent to which the SCHENGEN pathway uses components of MAMP signaling (as, for example, the requirement for SERK-family co-receptors, which has not been demonstrated yet for SGN3), but we propose that our findings on the SCHENGEN pathway function can already be of considerable interest for MAMP receptor signaling. The particular, restricted spatial overlap of receptor and downstream kinase has allowed us to visualize that, even if NADPH oxidases are non-localized (as in the case of RBOHD), ROS can be locally produced in a micrometer-scale region at the plasma membrane, through localized receptor stimulation. In MAMP receptor signaling, the non-localized nature of both MAMP receptor and NADPH oxidase would not have allowed to visualize this unexpected degree of spatial control. Yet, during actual microbial infections, highly localized receptor stimulation and ROS production might indeed occur and be relevant for the outcome of the immune response. In addition, while it is established that MAMP-stimulated ROS production is an important part of the immune response, its direct molecular downstream action is not well understood. Lignin production has long been associated with immune responses, as well as responses to cell wall damage, yet a direct molecular connection from MAMP perception to lignification has rarely been drawn (Chezem *et al*, 2017). It will be intriguing to investigate whether and how much MAMP-induced ROS production is actually used for lignification during defense and to which degree this explains the importance of ROS in plant defense responses.

The SCHENGEN pathway might not be limited to the regulation of lignified diffusion barriers, as it has been shown that it is also important in the formation of the embryonic cuticle. The peptide ligand used in this context has not been identified, and it remains unclear whether an equally precise barrier surveillance mechanism is also acting to ensure separation of endosperm and embryo during embryonic cuticle formation (Creff *et al*, 2019). In this context, the SCHENGEN pathway might be used to drive production and deposition of cutin instead of lignin and suberin as in the case of the endodermis. In the future, it will be fascinating to investigate whether

**Figure 7. Overview of the plasma membrane-based and nuclear branches of the SGN3 pathway.**

A, B (A) Schematic of the spatially restricted activation of ROS production at the plasma membrane after CIF stimulation (B) Comparison of the components of the cytoplasmic/nuclear signaling cascades induced by the flg22 bacterial pattern peptide (left) and the CIF peptides (right). Note that all of the signaling components in the two pathways belong to the same gene families, with the exception of the transcription factors. Components for which there is currently no direct experimental evidence are marked in gray, as are arrows indicating activation events that have not yet been experimentally established.

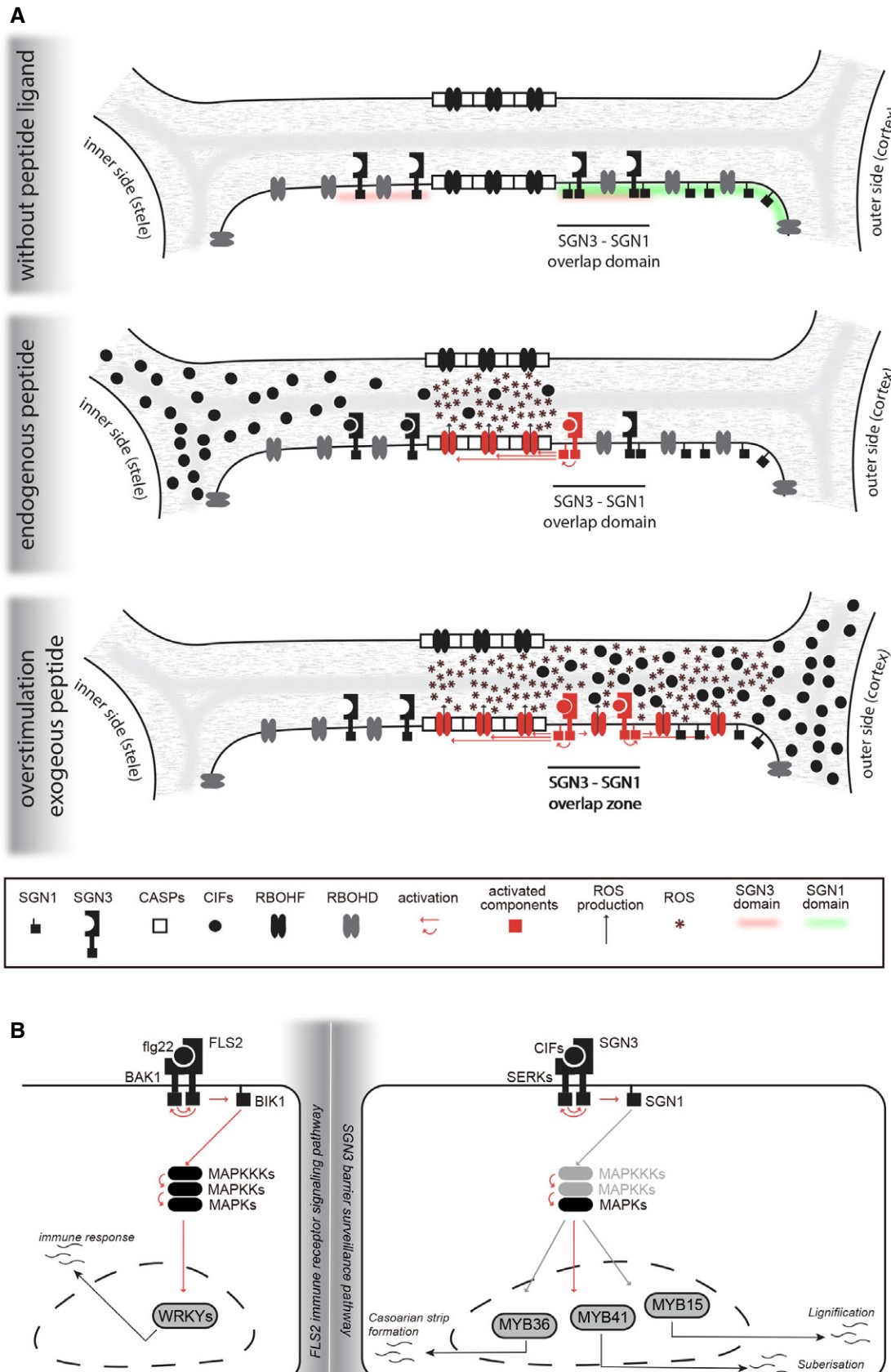

**Figure 7.**

the SCHENGEN pathway is of even broader developmental significance and to understand the molecular basis that enables its distinct, organ, and cell type-specific activities.

# Materials and Methods

### Plant material and growth conditions

For all experiments, *Arabidopsis thaliana* (ecotype Columbia) was used. The T-DNA-tagged lines, *sgn1-2* (SALK_055095C), and *sgn3-3* (SALK_043282) were obtained from NASC (Pfister *et al*, 2014; Alassimone *et al*, 2016). *rbohD dSpm*, *rbohF dSpm*, and *rbohD dSpm rbohF dSpm* are described in Tissier *et al* (1999); Torres *et al* (2002). *esb1-1* was kindly shared by Prof. David Salt's group (Baxter *et al*, 2009; Hosmani *et al*, 2013). The *cif1-2 cif2-2* double mutant was generated by CRISPR-Cas9 method (Fauser *et al*, 2014) using 5′- gctttggtttaggactggag -3′ as a protospacer sequence to target both CIF1 and CIF2 loci.

   *pCASP1::CASP1–GFP* marker line was crossed into *sgn1-2*, *sgn3-3*, or *cif1-2 cif2-2*. The *pCASP1::myrpalm SGN1* construct was independently transformed into each mutant background using the floral dip method (Clough & Bent, 1998). For observations and histological analysis, seeds were kept for 2 days at 4°C in the dark for stratification and then grown for 5 days at 22°C under 16-h light/8-h dark vertically on solid half-strength Murashige–Skoog (MS) medium. Cycloheximide (Sigma) and diphenyleneiodonium chloride (Sigma) were prepared as 50 mM water stock solution and as 20 mM DMSO stock solution, respectively.

### Plasmid construction

To generate myrpalm SGN1, GGCFSKK (5′- GGAGGATGCTTCTCTAAGAAG -3′) (Vermeer *et al*, 2004) was added to SGN1 cDNA right after the start codon by inverse PCR. This SGN1 variant was combined with pCASP1 and mCitrine coding sequence by LR clonase (Thermo Fisher Scientific). Coding sequences of RBOHF 2–383 a.a. and RBOHD 1–376 a.a. were inserted between BamHI and XhoI sites of a modified pET24(+) vector, in which GST (glutathione S-transferase) was added to the N-terminus. For constructing a GST-SGN3 kinase domain vector, a cDNA fragment coding for 899 a.a. to 1249 a.a. of SGN3 was fused into pDEST15 by Gateway LR reaction. TF-SGN1 was as described previously (Alassimone *et al*, 2016). Kinase-dead mutations in SGN1 (K134E) and SGN3 (K979E) were introduced by site-directed mutagenesis.

### Protein expression in bacteria and purification

Glutathione S-transferase or trigger factor (TF) proteins were expressed from pGex6p-1 or pCold-TF vectors in BL21 (DE3) CodonPlus RIL and purified in a glutathione sepharose column (Thermo Fisher Scientific) or a Co-Sepharose column (Clontech) according to the manufacturer's protocols. GST-RBOHD N-ter and GST-RBOHF N-ter were produced and purified as described in Drerup *et al* (2013). Bacterially expressed GST-SGN3 kinase domain (WT or kinase dead) in BL21 (DE3) CodonPlus RIL was purified with a glutathione sepharose column following the manufacture's protocol. For TF-SGN1 (WT or kinase dead), we followed the

method in the previous report (Alassimone *et al*, 2016). The buffer of all proteins was changed to 50 mM HEPES-KOH pH7.5 and 1 mM DTT with PD-10 or NAP-5. Purified kinase protein solutions were mixed with glycerol to 30% [v/v] final concentration and kept at −20°C until use.

### *In vitro* kinase assay

*In vitro* kinase assay was done as described previously (Alassimone *et al*, 2016) with small modifications. Purified 250 ng GST-SGN3 kinase domain was incubated with 250 ng TF-SGN1 or TF proteins in reaction buffer (50 mM HEPES-KOH pH7.5, 1 mM MnCl$_2$. 1 mM DTT, 1 mM ATP, 185,000 Bq [$\gamma$-$^{32}$P] ATP) at 30°C for 30 min. 250 ng TF-SGN1 and 1 μg GST or 500 ng GST-N-terminal cytoplasmic domains of RBOHD or F were treated in the same way as above. The reaction was stopped by adding 4 × LDS sample buffer (Invitrogen) and heating at 75°C for 10 min. The samples were separated on 4–12% gradient Nu-PAGE gels or 10% Nu-PAGE gels (Invitrogen). After drying the gels, signal was detected using Typhoon FLA7000 (GE Healthcare).

### Western blotting

Plants were grown in liquid MS medium supplemented with 0.5% sucrose with cell strainer for 5 days under long day (18-h light/6-h dark) condition after 2-day vernalization at 4°C. Hydroponically grown seedlings were transferred into fresh liquid MS medium containing 0.5% sucrose with or without 1 μM CIF2 peptide and incubated for 15 min. The seedlings were immediately frozen in liquid nitrogen and ground by TissueLyser II (Qiagen). Extraction buffer (20 mM Na phosphate pH 7.4, 150 mM NaCl, 1 mM EDTA, 0.1% Tween, 50 mM β glycerophosphate, 0.5 mM PMSF, 100 μM sodium orthovanadate, 5 mM Na fluoride, Complete cocktail) was added to the frozen samples. The samples were briefly mixed by vortex and centrifuged at 20,400 × *g* for 15 min at 4°C. The supernatant was transferred in new tubes and measured the concentration by Bradford method (Thermo Fisher Scientific). Equal amount of protein was loaded (20 μg/lane) and separated in 10% acrylamide gel (Eurogentec). After the electrophoresis, the separated proteins were transferred onto nitrocellulose membrane (Amersham, GE Healthcare) by XCell SureLock (Invitrogen) and stained with Ponceau S to show the loading control. The blot was incubated in blocking buffer (3% skim milk in TBS) for 1 h and probed with anti-phospho-p42, p44 antibody (1:1,000 dilution, Cell Signaling Technologies #9101) in 0.5% skim milk in TBS-0.1% Tween for overnight at 4°C. After washing the blot three times with TBS-0.1% Tween, the blot was treated with anti-rabbit secondary antibody (1:30,000 dilution, Agrisera) in 0.5% skim milk in TBS-0.1% Tween for 1 h and washed three times with TBS-0.1% Tween. Signals were detected on X-ray film with SuperSignal West Femto Kit (Thermo Scientific). For the MPK6 detection, the membrane was stripped (Pierce), incubated in blocking buffer for 1 h, and probed with anti-MPK6 antibody (1:10,000 dilution, Sigma) in blocking buffer for 1 h at room temperature. After washing the blot three times with TBS-0.1% Tween, the membrane was treated with anti-rabbit secondary antibody (1:30,000, Agrisera) in blocking buffer and washed three times. Signal was detected with SuperSignal West Femto kit (Thermo Scientific).

## Confocal microscopes

Confocal pictures were obtained using Leica SP8 or Zeiss LSM 880 confocal microscopes. The excitation and detection window were set to obtain signal as follows: When using Leica SP8 (excitation, detection window), GFP (488 nm, 500–550 nm), mCitrine, (514 nm, 518–560 nm), mCitrine/mCherry, (514 nm and 594 nm, 518–560 nm, 600–650 nm, sequential scan), when using Zeiss LSM880 (excitation, detection window), GFP (488 nm, 500–550 nm), Calcofluor White (405 nm, 425–475 nm), Basic Fuchsin (561 nm, 570–650 nm) and Fluorol Yellow (488 nm, 500–550 nm).

## PI assay

PI assay was done as described previously (Lee *et al*, 2013) with small changes. Seedlings were incubated in water containing 10 μg/ml PI for 10 min and transferred into fresh water. The number of endodermal cells was scored using a Zeiss LSM 700 confocal microscope (excitation 488 nm, SP640, split at 570 nm) from the onset of cell elongation (defined as endodermal cell length being more than two times than width in the median, longitudinal section) until PI could not penetrate into the stele.

## Scoring discontinuities in CS domain

CASP1-GFP signal was obtained by Leica SP8 as 1 μm step z-stack images from 5-day-old seedlings. After the images were projected as maximum projections, total lengths of CS domains were measured in maximum projections and the number of discontinuities in the domain was counted manually.

## Lignin and cell wall staining

ClearSee-adapted cell wall staining was performed as described in recent publications (Kurihara *et al*, 2015; Ursache *et al*, 2018). Briefly, 7–8 five-day-old seedlings were fixed in 3 ml 1 × PBS containing 4% paraformaldehyde for 1 h at room temperature in 12-well plates and washed twice with 3 ml 1 × PBS. Following fixation, the seedlings were cleared in 3 ml ClearSee solution under gentle shaking. After overnight clearing, the solution was exchanged to new ClearSee solution containing 0.2% Fuchsin and 0.1% Calcofluor White for lignin and cell wall staining, respectively. The dye solution was removed after overnight staining and rinsed once with fresh ClearSee solution. The samples were washed in new ClearSee solution for 30 min with gentle shaking and washed again in another fresh ClearSee solution for at least one overnight before observation.

## Methanol-based Fluorol Yellow staining of *Arabidopsis* root suberin

Vertically grown 5-day-old seedlings were incubated in methanol for 3 days at room temperature. The cleared seedlings were transferred to a freshly prepared solution of Fluorol Yellow 088 (0.01%, in methanol) and incubated for 1 h. The stained seedlings were rinsed shortly in methanol and transferred to a freshly prepared solution of aniline blue (0.5%, in methanol) for counterstaining. Finally, the seedlings were washed for 2–3 min in water and transferred to a chambered coverglass (Thermo Scientific), covered with a piece of

1% half-strength MS agar, and imaged using a Zeiss LSM 880 confocal microscope as described above.

## Detection of $H_2O_2$ production *in situ* using transmission electron microscopy

Visualization of $H_2O_2$ production around Casparian strip was done by cerium chloride method as described previously (Bestwick *et al*, 1997; Lee *et al*, 2013) with some modifications. Four-day-old *Arabidopsis* seedlings were transferred onto fresh 1/2 MS solid medium with or without 1 μM CIF2 and incubated for 24 h. After the peptide treatment, the seedlings were incubated in 50 mM MOPS pH7.2 containing 10 mM $CeCl_3$ for 30 min. After incubation with $CeCl_3$, seedlings were washed twice in MOPS buffer for 5 min and fixed in glutaraldehyde solution (EMS, Hatfield, PA) 2.5% in 100 mM phosphate buffer (pH 7.4) for 1 h at room temperature. Then, they were post-fixed in osmium tetroxide 1% (EMS) with 1.5% of potassium ferrocyanide (Sigma, St. Louis, MO) in phosphate buffer for 1 h at room temperature. Following that, the plants were rinsed twice in distilled water and dehydrated in ethanol solution (Sigma) at gradient concentrations (30% 40 min; 50% 40 min; 70% 40 min; two times (100% 1 h). This was followed by infiltration in Spurr resin (EMS) at gradient concentrations [Spurr 33% in ethanol, 4 h; Spurr 66% in ethanol, 4 h; Spurr two times (100% 8 h)] and finally polymerized for 48 h at 60°C in an oven. Ultrathin sections 50 nm thick were cut transversally at 1.3 ± 0.1 mm from the root tip, on a Leica Ultracut (Leica Microsystems GmbH, Vienna, Austria) and picked up on a copper slot grid 2 × 1 mm (EMS) coated with a polystyrene film (Sigma). Micrographs were taken with a transmission electron microscope FEI CM100 (FEI, Eindhoven, The Netherlands) at an acceleration voltage of 80 kV and 11,000 × magnification (pixel size of 1.851 nm, panoramic of 17 × 17 pictures), exposure time of 800 ms, with a TVIPS TemCamF416 digital camera (TVIPS GmbH, Gauting, Germany) using the software EM-MENU 4.0 (TVIPS GmbH, Gauting, Germany). All the pictures were taken using the same beam intensity, and panoramic aligned with the software IMOD (Kremer *et al*, 1996). For quantification, thresholding of cerium precipitate on pictures was scored using IMOD software (Kremer *et al*, 1996). Briefly, all pictures subjected to the quantification were normalized to one picture, a section from non-treated wild-type seedlings, by comparing the gray value of plastids in pericycle cells. After normalization, cell wall spaces from the beginning of the Casparian strip at the pericycle side until the cortex corner between endodermal cells were selected and one identical threshold setting was applied to all pictures in order to highlight signals around Casparian strips. The values were shown as a percentage of thresholded pixels to selected area.

## ROS production assay in HEK cells

Measurements of RBOHD and RBOHF activity were performed as described in Han *et al* (2019) with some modifications. RBOHD or RBOHF was transiently expressed in HEK293T cells with or without co-expression of SGN1 or myrpalm SGN1. HEK293T cells were cultivated at 5% $CO_2$ and 37°C in Dulbecco's modified Eagle's medium (DMEM) mixture F-12HAM (Sigma) enriched with 10% fetal bovine serum. Before transfection, lysine-coated white 96-well plates were inoculated with HEK293T cells and incubated for 24 h. For the transient transfection of the cells, GeneJuice transfection reagent

(Novagen) was used according to the manufacturer's guidelines. The transfected expression cassettes were cloned into modified pEF1 vectors (Drerup *et al*, 2013). Each well was transfected with 110 ng of plasmid mixture (50 ng pEF1-RBOH; 30 ng of each effector; empty pEF1 vector to ensure equal loading). Transfected cells were incubated for 48 h and subsequently measured in a buffer consisting of Hank's Balanced Salt Solution (Gibco) with 62 μM L-012 and 60 μg/ml HRP. To stimulate SGN1, calyculin A, a phosphatase inhibitor, was added at the final concentration of 0.1 μM directly before the start of the measurements. ROS production was measured in a LB 943 Mithras² (Berthold) microplate reader and is presented as relative luminescence units s$^{-1}$ (RLU s$^{-1}$) in graphs. For all experiments, the values were gained from 6 wells measured in parallel and average values were plotted with SD. Experiments have been repeated three times.

## Sample preparation for RNA-seq experiments

Wild-type, *sgn3-3,* and *cif1-2 cif2-2* were grown on solid half MS medium with mesh for 5 days and transferred onto fresh half MS medium with or without 100 nM CIF2. After 30-, 120-, or 480-min incubation, aerial parts were cut off and whole roots were collected. Samples were immediately frozen in liquid nitrogen, and RNA was extracted using a TRIzol-adapted ReliaPrep RNA extraction kit (Promega).

## RNA-seq library preparation and sequencing

RNA-seq libraries were prepared as described in Jan *et al* (2019). In brief, RNA quality was assessed on a Fragment Analyzer (Advanced Analytical Technologies, Inc., Ankeny, IA, USA). RNA-seq libraries were prepared using 1,000 ng of total RNA and the Illumina TruSeq Stranded mRNA reagents (Illumina; San Diego, California, USA) on a Sciclone liquid handling robot (PerkinElmer; Waltham, Massachusetts, USA) using a PerkinElmer-developed automated script. Cluster generation was performed with the resulting libraries using the Illumina TruSeq SR Cluster Kit v4 reagents and sequenced on the Illumina HiSeq 2500 using TruSeq SBS Kit v4 reagents. Sequencing data were processed using the Illumina Pipeline Software version 2.20.

## RNA-seq data processing and analysis

Data processing was performed by the Lausanne Genomic Technologies Facility using their in-house RNA-seq pipeline. It includes purity-filtered read trimming for adapters and low-quality sequences with Cutadapt (v. 1.8) (Martin, 2011). Removal of reads matching ribosomal RNA sequences was done with fastq_screen (v. 0.11.1), followed by low complexity read filtering with reaper (v. 15-065) (Davis *et al*, 20132013). Cleaned reads were aligned against *Arabidopsis thaliana* TAIR10 genome using STAR (v. 2.5.3a) (Dobin *et al*, 2013). Read counts were obtained per gene locus with htseq-count (v. 0.9.1) (Anders *et al*, 2015) using A. thaliana TAIR10 Ensembl 39 gene annotation. The quality of the data alignment was evaluated with RSeQC (v. 2.3.7; Wang *et al*, 2012).

Gene-level statistical analysis was performed in R (R version 3.4.3). Filtering of genes with low counts was achieved based on the rule of 1 count per million (cpm) in at least 1 sample. TMM normalization was applied for library sizes scaling followed by log-transformation into counts per million or CPM (EdgeR package version 3.20.8; Robinson *et al*, 2010). PCA was computed using normalized values corrected for batch effect using limma function removeBatchEffect.

Differential expression was computed with limma-trend approach (Ritchie *et al*, 2015) by fitting all samples into one linear model. The batch factor was added to model matrix.
Pairwise comparisons treated vs untreated per time point were assessed using moderated t-tests. The adjusted *P*-value is computed by the Benjamini–Hochberg method, controlling for false discovery rate (FDR or adj.*P*.Val).
Differential expression of untreated mutant vs wt per time point was assessed using moderated *F*-test and post hoc classification. The adjusted *P*-value is computed by the Benjamini–Hochberg method, controlling for false discovery rate (FDR or adj.*P*.Val).
Differential expression of treated vs untreated over time paired by genotype was assessed using moderated F-test and post hoc classification. The adjusted *P*-value is computed by the Benjamini–Hochberg method, controlling for false discovery rate (FDR or adj.*P*.Val).
Interaction between time and treatment (paired by genotype, excluding SGN3 of the model) was assessed using moderated *F*-test. The adjusted *P*-value is computed by the Benjamini–Hochberg method, controlling for false discovery rate (FDR or adj.*P*.Val).
Time effect in untreated conditions (paired by genotype, excluding SGN3 of the model) was assessed using moderated *F*-test. The adjusted *P*-value is computed by the Benjamini–Hochberg method, controlling for false discovery rate (FDR or adj.*P*.Val).

Genes were considered significant in further analysis if the adjusted *P*-value was equal or below 0.05 and the log$_2$ fold change was ≥ 1. Heatmaps were constructed using the package ComplexHeatmap (v1.99.4, Pearson distance for row clustering; Gu *et al*, 2016). Genes were clustered using kmeans (factoextra v1.0.5, https://github.com/kassambara/factoextra) using a non-supervised approach resulting in 3 suggested clusters. We found that 4 inferred clusters split one gene cluster in a more sensible way and adjusted the number of clusters accordingly to four. GO analysis was conducted using the package topGO (v. 2.34.0, weight01 algorithm; Alexa & Rahnenfuhrer, 2019). GO annotations were obtained through Biomart (version 2.40.0, Ensembl Plants release 43—April 2019; Durinck *et al*, 2009). Genes for pathway-specific heatmaps were obtained from the corresponding GO term through the Ensembl Plant database.

## qPCR analysis

For the CIF2 peptide treatment, 5-day-old seedlings grown on half MS were moved to fresh 1/2 MS medium in the absence or presence of 100 nM CIF2 and incubated for 30 or 120 min. Otherwise, the plants were grown with mesh. Only root parts (around 100 mg) were collected at each time point, and total RNA was extracted using a TRIzol-adapted ReliaPrep RNA Tissue Miniprep Kit (Promega). Reverse transcription was carried out with PrimeScript RT Master Mix (Takara). All steps were done as indicated in the manufacturer's protocols. The qPCR was performed on an Applied Biosystems QuantStudio3 thermocycler using a MESA BLUE SYBR Green kit (Eurogentech). All transcripts are normalized to *Clathrin adaptor complexes medium subunit family protein* (*AT4G24550*) expression. All primer sets are indicated in Table EV3.

**Statistical analysis for experiments**

All statistic-related analysis was done with R software (R Core Team, 2013) (https://www.r-project.org/). For multi-comparison analysis, one-way ANOVA was carried out and Tukey's test was subsequently performed.

## Data availability

All data to support the conclusions of this manuscript are included in the main text and supplementary materials. The full RNA-seq dataset was deposited in GEO (accession GSE144182, https://www.ncbi.nlm.nih.gov/geo/query/acc.cgi?acc=GSE144182).

**Expanded View** for this article is available online.

## Acknowledgements
We thank the Central Imaging Facility (CIF), Genome Technology Facility (GTF), particularly Sandra Calderon, and Electron Microscopy facility (EMF) of the University of Lausanne for expert technical support. We also thank Hiroko Uchida for expert graphical support. This work was supported by funds from an ERC Consolidator Grant (GA-N°: 616228—ENDOFUN) and two SNSF grants (CRSII3_136278 and 31003A_156261) to N.G., a Federation of European Biochemical Sciences Postdoctoral Long-Term Fellowship to P.M., an EMBO Long-term Postdoctoral Fellowship to R.U. and M.B., an Overseas Research Fellowship from JSPS to S.F., a Marie Curie Postdoctoral Fellowship to T.G.A., a fellowship of the Fundación Alfonso Martín Escudero to V.G.D, and a DFG grant (Ku931/14-1) to J.K.

## Author contributions
SF and NG conceived the project. SF, TGA, and NG designed the experiments. SF, DDB, KHE, PK, TGA, VDT, AP, PM, RU, VGD, MB, and AC performed the experimental work. DDB, TGA, ES-S, and JD performed image quantification and RNA-seq analysis. SF and NG wrote the manuscript. GI, JK, and all other authors revised the manuscript and were involved in the discussion of the work.

## Conflict of interest
The authors declare that they have no conflict of interest.

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
