## [Review Process File · The EMBO Journal]

SCHENGEN receptor module drives localized ROS production and lignification in plant roots

Satoshi Fujita, Damien De Bellis, Kai H. Edel, Philipp Köster, Tonni Grube Andersen, Emanuel Schmid-Siegert, Valérie Dénervaud Tendon, Alexandre Pfister, Peter Marhavý, Robertas Ursache, Verónica G. Doblás, Marie Barberon, Jean Daraspe, Audrey Creff, Gwyneth Ingram, Jörg Kudla and Niko Geldner

Review timeline:

Submission date:	3rd Nov 2019	Editorial Decision:
Revision received:	17th Jan 2020	
Editorial Decision:	12th Feb 2020	
Revision received:	21st Feb 2020	
Accepted:	26th Feb 2020	

Editor: Ieva Gailite

Transaction Report:

1st Editorial Decision

12th Dec 2019

Thank you for submitting your manuscript for consideration by the EMBO Journal. We have now received three referee reports on your manuscript, which are included below for your information.

As you will see from the comments, all reviewers appreciate the work and the quality of the data and recommend publication of the manuscript after a minor revision. Given these positive evaluations from three experts of the field, I would like to invite you to submit a revised version of your manuscript in which you address the issues raised in reviewers' reports.

REFeree REPORTS:

Referee #1:

In this manuscript, Satoshi Fujita and co-workers from the Geldner (and other) lab(s) present a major scientific advance by demonstrating how spatial control of lignification of a special cell wall domain, the Casparian Strip that serves as a diffusion barrier in the endodermal cell wall, can be directed at sub-micrometer resolution.

The genes involved in this process had previously been identified by the authors. Nevertheless, here, the authors have designed a cleverly thought-out, outstanding combination of genetic, biochemical and cell biological experiments largely conducted in planta but further supported by direct in vitro kinase activity assays as well as Reactive Oxygen Species (ROS) production experiments in the heterologous human HEK293T cells system. The combination of these experiments allows the authors to address the molecular functions and interactions of the SCHENGEN3 (SGN3)-SGN1 kinase relay system, demonstrating their direct interaction as well as the novel identification of the RBOHF and RBHOD NADPH oxidases as direct targets for activation through SGN1 kinase dependent phosphorylation. Membrane-localised myrpalmsGN1 is further demonstrated to strongly

stimulate ROS production by RBOHF and RBHOD when all components are heterologously expressed in HEK293T cells.

The highly localised ROS production at the very small site of overlapping SGN3-SGN1 plasma membrane localisation in planta is then analysed using a ROS detection assay at the electron microscopic level in plants. This demonstrates, factually at a resolution of down to around 100 nm, where ROS is produced locally in mutants of the receptor kinase SGN3, the downstream kinase SGN1 and mutants for its direct substrates the RBOHD as well as RBOHDF. Both local endogenous ROS production as well as strong ROS production after exogenous application of the CIF2 peptide ligand for SGN3 is completely abolished in the *sgn3* mutant as well as in the *rboh*d *rboh*f double mutant and largely abolished in the *sgn1* mutant. These findings demonstrate demonstrating that this peptide ligand-SCHENGEN kinase relay module is responsible for highly localised ROS production via RBOHF and RBOHD required for lignin polymerization at a sub-micrometer level. These findings at such are novel and conceptionally unprecedented.

Since the highly localised ROS and lignin production are not the only read-outs of the SGN3 pathway, which also is needed for continuous CASPARIAN STRIP DOMAIN PROTEIN (CASP) protein accumulation as well as suberin synthesis, the authors further analysed transcriptional changes after application of the CIF2 peptide (in the *cif1 cif2* double mutant) in presence of cycloheximide to identify fast and direct CIF2 pathway response genes. This identified strong upregulation of PEROXIDASE15/49 genes encoding enzymes likely catalyzing the last step in ROS-dependent monolignol radical formation, similarly laccases and genes involved in suberin biosynthesis as well as the MYB15 transcription factor required for microbe-assisted molecular pattern (MAMP) perception, breaking new ground for exciting future research.

The main findings demonstrating the local action of a kinase relay system on highly localised ROS production and the resulting lignification are unprecedented in the plant research field. Because they decipher an (almost) complete signal transduction pathway at the genetic, molecular and sub-cellular level leading to localised cell wall modification they will also be of conceptual interest to molecular, cell and developmental biologists working on non-plant systems.

The experiments are conducted carefully, the figures are highly informative and well displayed. The manuscript is of interest to a general readership, the main text is written for such a readership and spotless, while a number of spelling and language mistakes can be found in the Materials and Methods/Supplementary Figures that should be proof-read prior to publication.

The manuscript after minor modifications (see below) will be highly suitable for publication in a journal addressing a general readership of molecular, cell and developmental biologists such as The EMBO Journal.

Minor issues:

- 1) Proofread the Material and Methods and the Supplementary Material for spelling/language errors. There are a number of them such as "thresfold" in Fig. S4 etc.
- 2) What is $n = 12$ and $n = 10$ in Figs. 2A and 2B, respectively? The number of roots? Please add.
- 3) How many roots and cells were analysed in the experiments in Fig. 3 and how many of them showed the indicated phenotypes. Please, amend.
- 4) How many roots and cells were analysed in the experiments in Figs. &A-C and how many of them showed the indicated phenotypes. Please, amend.
- 5) While I do not think that this experiment should be conducted within the scope of this manuscript, I wondered as to whether SGN3 has also be tested for potentially providing a substrate for SGN1. As a very strong signal needs to be created at a very small domain of overlapping localisation, I wondered as to whether SGN1 could directly positively feed back on SGN3 molecules to create local signal amplification.

6) If one considers the NADPH oxidase the last part of the signal transduction chain one could say the authors have delineated a complete signal transduction pathway. Still, NADPH oxidase action should be followed by superoxide dismutase and then peroxidase activity. I, therefore, wondered as to whether, similar to the two peroxidases, some superoxide dismutase transcripts came up in the transcriptional profiling analyses conducted by the authors.

Referee #2:

The manuscript by Fujita et al is in general an interesting set of observations detailing the kinase cascade acting to localise ROS in casparian strip production. This is a large body of work but authors seem to overinterpret their data which reduces the value. At times there appears an attempt to make the manuscript more fascinating instead of plainspeak! A lot of non-scientific terminology like DIRECT, FASCINATING and so on. I have the following comments:

Line 33: remove the word full. In biology nobody knows what identification of full is. It is a non-scientific term!

Line 34: remove the word direct. I don't know what this would mean! I can imagine a transcription factor binding to site and exerting a direct control but kinase cascade? Hmmm!

Line 35: This last sentence is not possible to understand and feels like an attempt to make the manuscript fancy. Please simplify

Line 42: Refrain from using terms such as "most critical" everything is critical. Just use a simple term like crucial or so. We get it that study of ROS is important.

Line 55: it is weird way to put that peroxidase are "targets" of ROS when the actual reactive oxygen species is used by peroxidases as a substrate (by analogy that would mean that all enzymes in a chemical reaction are targets of its substrates). It would be one thing if this ion would modify the peroxidase itself then I would agree that it is a target! Reformulate please.

Line 88: Formulation that pathway allows endodermal cell layer to probe is also weird reformulate by eliminating endodermal cell layer. The way it is written, it gives endodermal layer decision making and thinking capacity!

Line 96: Again remove the word full.

Results: Why was the SGN1 variants expressed from endodermal promoters instead of native promoters? Couldn't this create artifacts?

Couldn't an alternative explanation of sgn1 phenotype be that there is a complex of some sort and interaction between sgn1 and sgn3 and in absence of sgn1, this complex is less effective? In which case sgn1 is not downstream of sgn3. This is what happens in enzyme complexes so I am not sure the formulation is correct here.

Line 217: sgn3 has no ROS and sgn1 has reduced ROS that cannot be enhanced. Based on this I would not conclude that highly localized ROS was dependent on SGN3/SGN1 module. For implying localized ROS dependence on this module you would need preferably a loss of mutation in a gene which results in ROS is delocalized. So what you have here is spatial correlation between SGN3/SGN1 and ROS but mutant analysis does not prove this point.

About SGN1 phosphorylation of RBOs. Remove the word direct and I also would soften the point of direct phosphorylation. All experiments are essentially indirect. To make the point of directness (used annoying often without sufficient evidence) you would need to demonstrate in addition to these experiments lack of phosphorylation of RBOs in SGN1 mutant background in planta!

Finally I think the entire chapter on gene expression is rather irrelevant to the story and simply complicates it!

Referee #3:

- general summary and opinion about the principle significance of the study, its questions and findings

These last years, the Geldner's lab, following a pioneering mutant screen, unravelled the Schengen mutants, affected in the deposition of the Casparian strip in plants. Through the characterization of these mutants and their corresponding proteins, emerged progressively a picture highlighting the critical importance of membrane micro-domains in the fine regulation of cell wall polymer deposition in plants, using the Casparian strip as a main model. This previous work pointed that SGN1, 3, and the CIF2 peptides may act in similar signalling pathways. Here, the authors provide a mechanistic evidence that SGN3 and SGN1 act in a similar signalling cascade, and that SGN3 is the most critical initiator of this pathway for precisely localized deposition of lignin in the Casparian strip through. The work elegantly reconciles the genetic and cellular evidence with kinase assays, highlighting the phosphorylation signalling hierarchy of SGN3 over downstream SGN1, leading to the subsequent activation of NOX proteins producing ROS, a necessary step for lignin polymerization. It allows a more comprehensive view of the Schengen pathway at signalling level, and will benefit the community when addressing the role of spatialization and signalling mechanisms at play during the critical steps of cell wall deposition during plant development.

However elegant is the study, we would like the authors to address the following points:

- specific major concerns essential to be addressed to support the conclusions
- minor concerns that should be addressed
- We wonder why the authors didn't assess whole RNA profiling after CIF treatments in *sgn1* background, as they are undertaking such analysis in the *sgn3* background and assessing the expression some specific genes in the *sgn1* background. This part of the manuscript is interesting to assess what genetic responses are triggered by the CIF signalling, and reaffirming that SGN3 is indeed the main receptor if CIF responses. However, assessing such responses in a more holistic manner in a *sgn1* background would allow to discriminate more precisely the genetic responses more specifically attributed to the SGN1 subsequent signalling pathway. Such experiment would be relevant to discuss this aspect.
- The authors mentioned that probable SGN1 homologs can partially compensate for its absence in a *sgn1* background. Do they have any genetic evidence for this? (higher order mutant) If yes, similarly to the question raised above, RNA profiling in such background would be interesting to assess.
- any additional non-essential suggestions for improving the study (which will be at the author's/editor's discretion)
- Can the authors explain the presence of possible ROS production in the *sgn3* background after CIF2 addition? (visible as black dots on the bottom left of Fig. 4D, *sgn3*; slightly statistically different when considering only *sgn3* background in Fig. 4E) Indeed, If SGN3 CIF interaction is necessary to trigger ROS production through NOX proteins, ROS shouldn't be detected.
- A fluorescence quantification of the outer vs inner fluorescence of SGN1-Cit and myrpalm-SGN1-CIT would be interesting in order to appreciate in a more quantitative manner the broader localization of SGN1 after myrpalm addition (eg fluo ratio outer vs inner side in the 2 conditions).
- A graphical abstract might be relevant to offer a more graphical overview of the main MS findings for non specialists.
- A more detailed legend on the 2 panels of each kinase assays (Fig2. and Fig.5) would be appreciated to ease the reader understanding of these experiments.
- p4, l136: We believe the authors made a typo and refer to Fig. 1C when it should refer to Fig. 1D.

1st Revision - authors' response

17th Jan 2020

POINT-BY-POINT REPLY to Reviewer comments – Fujita et al.

Referee #1:

In this manuscript, Satoshi Fujita and co-workers from the Geldner (and other) lab(s) present a major scientific advance by demonstrating how spatial control of lignification of a special cell wall domain, the Casparian Strip that serves as a diffusion barrier in the endodermal cell wall, can be directed at sub-micrometer resolution.

The genes involved in this process had previously been identified by the authors. Nevertheless, here, the authors have designed a cleverly thought-out, outstanding combination of genetic, biochemical and cell biological experiments largely conducted in planta but further supported by direct in vitro kinase activity assays as well as Reactive Oxygen Species (ROS) production experiments in the heterologous human HEK293T cells system. The combination of these experiments allows the authors to address the molecular functions and interactions of the SCHENGEN3 (SGN3)-SGN1 kinase relay system, demonstrating their direct interaction as well as the novel identification of the RBOHF and RBHOD NADPH oxidases as direct targets for activation through SGN1 kinase dependent phosphorylation. Membrane-localised myrpalmSGN1 is further demonstrated to strongly stimulate ROS production by RBOHF and RBHOD when all components are heterologously expressed in HEK293T cells.

The highly localised ROS production at the very small site of overlapping SGN3-SGN1 plasma membrane localisation in planta is then analysed using a ROS detection assay at the electron microscopic level in plants. This demonstrates, factually at a resolution of down to around 100 nm, where ROS is produced locally in mutants of the receptor kinase SGN3, the downstream kinase SGN1 and mutants for its direct substrates the RBOHD as well as RBOHDF. Both local endogenous ROS production as well as strong ROS production after exogenous application of the CIF2 peptide ligand for SGN3 is completely abolished in the *sgn3* mutant as well as in the *rboh*d *rboh*f double mutant and largely abolished in the *sgn1* mutant. These findings demonstrate demonstrating that this peptide ligand-SCHENGEN kinase relay module is responsible for highly localised ROS production via RBOHF and RBOHD required for lignin polymerization at a sub-micrometer level. These findings at such are novel and conceptually unprecedented.

Since the highly localised ROS and lignin production are not the only read-outs of the SGN3 pathway, which also is needed for continuous CASPARIAN STRIP DOMAIN PROTEIN (CASP) protein accumulation as well as suberin synthesis, the authors further analysed transcriptional changes after application of the CIF2 peptide (in the *cif1 cif2* double mutant) in presence of cycloheximide to identify fast and direct CIF2 pathway response genes. This identified strong upregulation of PEROXIDASE15/49 genes encoding enzymes likely catalyzing the last step in ROS-dependent monolignol radical formation, similarly laccases and genes involved in suberin biosynthesis as well as the MYB15 transcription factor required for microbe-assisted molecular pattern (MAMP) perception, breaking new ground for exciting future research.

The main findings demonstrating the local action of a kinase relay system on highly localised ROS production and the resulting lignification are unprecedented in the plant research field. Because they decipher an (almost) complete signal transduction pathway at the genetic, molecular and sub-cellular level leading to localised cell wall modification they will also be of conceptual interest to molecular, cell and developmental biologists working on non-plant systems.

The experiments are conducted carefully, the figures are highly informative and well displayed. The manuscript is of interest to a general readership, the main text is written for such a readership and spotless, while a number of spelling and language mistakes can be found in the Materials and Methods/Supplementary Figures that should be proof-read prior to publication.

The manuscript after minor modifications (see below) will be highly suitable for publication in a journal addressing a general readership of molecular, cell and developmental biologists such as The EMBO Journal.

Minor issues:

- 1) Proofread the Material and Methods and the Supplementary Material for spelling/language errors.

There are a number of them such as "thresfold" in Fig. S4 etc.

REPLY: Thank you for pointing out our mistakes. We have fixed the misspelling "thresfold" to threshold in FigS4.

2) What is $n = 12$ and $n = 10$ in Figs. 2A and 2B, respectively? The number of roots? Please add.

REPLY: Thank you for the comments. n -values indicate the number of the roots observed. We changed the figure legend accordingly.

3) How many roots and cells were analysed in the experiments in Fig. 3 and how many of them showed the indicated phenotypes. Please, amend.

REPLY: Thank you for pointing out this issue. We agree that adding numeric information would improve the quality of the data. Regarding to Figure 3A, we checked 2 spots from 10 roots for each transgenic line. Moreover, as we mentioned in the text, these localization patterns are consistent with the data in Lee et al. (2013). For Figure 3B, we observed 2 spots from 12 roots for each condition. ALL cells we observed showed similar patterns to the representative pictures that we presented in the figure, which is why we did not find it useful to quantify this effect. We added this information in the figure legend of Fig. 3

4) How many roots and cells were analysed in the experiments in Figs. &A-C and how many of them showed the indicated phenotypes. Please, amend.

REPLY: Again, we agree that adding numeric information would make our data more reliable. We assume that this reviewer made typo but most likely it would be for Fig1 or Fig4. We added the information requested by reviewer 1 in the legends for figure 1CD and Fig4A-C. We obtained almost identical patterns in ALL roots.

Since we completely agree the point of reviewer1, we added the numeric information in similar figures, such as FigS1 or FigS3

5) While I do not think that this experiment should be conducted within the scope of this manuscript, I wondered as to whether SGN3 has also be tested for potentially providing a substrate for SGN1. As a very strong signal needs to be created at a very small domain of overlapping localisation, I wondered as to whether SGN1 could directly positively feed back on SGN3 molecules to create local signal amplification.

REPLY: We haven't tested this and we agree that it would be interesting. To our knowledge, it has not been reported for an RLCKVII kinase to phosphorylate its own receptor in a feedback loop, but this is clearly a possibility to be explored. However, we would prefer to wait with further experiments in this direction until we have obtained a better idea of the co-receptor(s) of SGN3.

6) If one considers the NADPH oxidase the last part of the signal transduction chain one could say the authors have delineated a complete signal transduction pathway. Still, NADPH oxidase action should be followed by superoxide dismutase and then peroxidase activity. I, therefore, wondered as to whether, similar to the two peroxidases, some superoxide dismutase transcripts came up in the transcriptional profiling analyses conducted by the authors.

REPLY: There is indeed the possibility that superoxide dismutases (SODs) are required and we have previously undertaken some experiments in this regard (Yuree Lee and Aurélie Emonet, *unpublished*). Of all annotated SODs, we only found one to be localized to the apoplast, but we did not observe any lignification phenotype in the knock-out mutant of this SOD. This would mean that still unknown SODs are used for dismutation in the apoplast or that the specific apoplastic conditions lead to fast-enough, spontaneous dismutation of superoxide to H₂O₂. This has repeatedly been discussed as a possibility in the literature.

Referee #2:

The manuscript by Fujita et al is in general an interesting set of observations detailing the kinase

cascade acting to localise ROS in casparian strip production. This is a large body of work but authors seem to overinterpret their data which reduces the value. At times there appears an attempt to make the manuscript more fascinating instead of plainspeak! A lot of non-scientific terminology like DIRECT, FASCINATING and so on. I have the following comments:

Line 33: remove the word full. In biology nobody knows what identification of full is. It is a non-scientific term!

REPLY: We removed the term- but we disagree with the reviewer. In the context of signal transduction research, it is a very useful, operative term in order to indicate that there is no additional kinase, or other, component required in order to account for the activation of downstream targets, in our case the NADPH oxidase. We demonstrate that SGN3 phosphorylates SGN1 *in vitro* and that SGN1 phosphorylates RBOHF and activates it in a heterologous system. Thus, we have established a “full” circuit, a term which we wouldn’t have used if we had been unable to demonstrate direct phosphorylation of SGN1 by SGN3, for example. In this case we would have had to postulate an additional “component X” that mediates activation of SGN1 by SGN3.

Line 34: remove the word direct. I don't know what this would mean! I can imagine a transcription factor binding to site and exerting a direct control but kinase cascade? Hmmm!

REPLY: We used the term “direct” here precisely to indicate that the CIF ligands do not spatially control lignification by causing transcriptional activation of peroxidases that then somehow find their way to the right place (or are activated in the right place) by unknown mechanisms. . . rather, our data allows to draw a model in which SGN3 activates SGN1, which activates RBOHF, which produces ROS, which enables peroxidases to polymerise lignin – a “direct” regulation, using pre-existing plasma membrane/apoplast localized components.

Line 35: This last sentence is not possible to understand and feels like an attempt to make the manuscript fancy. Please simplify

REPLY: We have provided an additional model figure (Figure 7), which hopefully makes this point of “intersection of more broadly localized components” very clear. We think it is a very good summary of our results.

Line 42: Refrain from using terms such as "most critical" everything is critical. Just use a simple term like crucial or so. We get it that study of ROS is important.

REPLY: We replaced it with the term “important”. Not everything is critical, there are functions that, when knocked-out, strongly interfere with normal development, viability and reproduction, even under lab conditions (“critical”), others have only minor, quantitative effects under the same conditions (“non-critical”).

Line 55: it is weird way to put that peroxidase are "targets" of ROS when the actual reactive oxygen species is used by peroxidases as a substrate (by analogy that would mean that all enzymes in a chemical reaction are targets of its substrates). It would be one thing if this ion would modify the peroxidase itself then I would agree that it is a target! Reformulate please.

REPLY: We agree, we now talk about “ROS action” and immediately mention that ROS is a co-substrate.

Line 88: Formulation that pathway allows endodermal cell layer to probe is also weird reformulate by eliminating endodermal cell layer. The way it is written, it gives endodermal layer decision making and thinking capacity!

REPLY: We changed this sentence, it now reads: “This pathway would therefore provide a mechanism that allows to probe the tissue-wide integrity of an extracellular diffusion barrier and to respond to barrier defects through compensatory over-lignification.”

Line 96: Again remove the word full.

REPLY: We removed the term.

Results: Why was the SGN1 variants expressed from endodermal promoters instead of native promoters? Couldn't this create artifacts?

REPLY: SGN1 promoter is very similarly expressed than the utilized CASP1 promoter (See Fig. 3A-D in Alassimone, Fujita et al., 2016), but it is weaker. We used the CASP1 promoter, because we wanted to be maximise our chances to observe inner SGN1 localisation (in case it would be weak) and to observe effects of this mis-localisation. However, we have done every control that we could conceive in order to exclude that this overexpression leads to artifacts. We have shown

expression of wild-type version under the same promoter does not lead to mislocalisation of SGN1 or ectopic lignin formation in *wild-type* (1C, Col) and fully complements the *sgn1* mutant (1C, *sgn1*). Moreover, we have shown that mis-localised SGN1 leads to ectopic lignification only when SGN3 is present, excluding potential, constitutive activity due to overexpression and/or mislocalisation (compare 1D Col with 1D *sgn3*).

Couldn't an alternative explanation of *sgn1* phenotype be that there is a complex of some sort and interaction between *sgn1* and *sgn3* and in absence of *sgn1*, this complex is less effective? In which case *sgn1* is not downstream of *sgn3*. This is what happens in enzyme complexes so I am not sure the formulation is correct here.

REPLY: If we had no molecular knowledge of the components whatsoever, we would agree with the reviewer. However, SGN3 has been demonstrated to physically bind to CIF2 and to be fully required for its *in vivo* activity. This makes sense with its sequence homology to LRR-RLK, of which many have shown to be receptors. SGN1 is homologous to RLCKs of subfamily VII, i.e. cytoplasmic kinases that have been shown repeatedly to be phosphorylated and activated by LRR RLK and to be necessary for signal transduction (see FLS2 and BIK1, for example). In this context, it is very appropriate, in our opinion, to talk about SGN1 being “downstream” of SGN3, since the initial event (ligand binding) must occur in the apoplast and is mediated by SGN3, while SGN1 is required for transduction of the signal, but is on the cytosolic site and phosphorylated by SGN3. Thus, it is – in signal transduction terms – downstream of the receptor.

Line 217: *sgn3* has no ROS and *sgn1* has reduced ROS that cannot be enhanced. Based on this I would not conclude that highly localized ROS was dependent on SGN3/SGN1 module. For implying localized ROS dependence on this module you would need preferably a loss of mutation in a gene which results in ROS is delocalized. So, what you have here is spatial correlation between SGN3/SGN1 and ROS but mutant analysis does not prove this point.

REPLY: I think we have indeed different understanding of the meaning of terms. We are very sorry about this. We would maintain that if a localized ROS production is observed in wild-type and abolished in a mutant (as we observe here) than this localized ROS production “is dependent on” or “requires” the gene that has been mutated. In my understanding of genetic terminology “required” or “dependent on” does not mean that the mechanism whereby localization is achieved is explained by this gene. A gene whose loss-of-function delocalizes ROS would indeed partially explain the localization, but not the ROS production in the first place (if such a gene exists). The striking spatial co-incidence of SGN3 SGN1 localisation overlap and ROS production indeed only suggests, but does not demonstrate, that localization of the module in itself explains the localization of ROS production. We extended the sentence and added the term “suggest” to clarify this.

About SGN1 phosphorylation of RBOs. Remove the word direct and I also would soften the point of direct phosphorylation. All experiments are essentially indirect. To make the point of directness (used annoying often without sufficient evidence) you would need to demonstrate in addition to these experiments lack of phosphorylation of RBOs in SGN1 mutant background in planta!

REPLY: Again, let us attempt to explain our understanding of the term “direct”. We are using it only if there is evidence of physical interaction between two components AND if the components can actually encounter each other *in vivo* AND if there is *in vivo* evidence that they actually do regulate each other. In the case of SGN1 we have all of the above: (i) *in vitro* phosphorylation with recombinant proteins in the absence of any other components, (ii) demonstration that SGN1 is capable of activating RBOHF in a heterologous system (mammalian cell cultures), (iii) endogenous expression of both components in endodermal cells and localization to the same compartment (cytosolic face of the PM) and (iv) demonstration that RBOHF activity *in vivo* is largely reduced in the absence of SGN1.

Finally I think the entire chapter on gene expression is rather irrelevant to the story and simply complicates it!

REPLY: We are very sorry, but we disagree with the reviewer. We invested considerable effort and money to obtain a well-designed and properly-analysed RNA response profile after ligand stimulation. For the CIF – SGN3 pathway only very little was known about its effect on transcriptional regulation and no transcriptomic analysis had been done. How such a dataset can be considered irrelevant is hard for us to understand. Moreover, we think it is relevant to the story at hand: In this and previous papers, we have provided a number of cellular differentiation effects of

the CIF pathway, namely enhanced and delocalized lignification, enhanced suberisation and enhanced production and accumulation of CASPs. For all three of these effects we can see underlying transcriptional changes, i.e. upregulation of peroxidases, upregulation of suberin biosynthetic enzymes, as well as upregulation of CASPs. Our gene expression set therefore provides an important element of explanation for the observed cellular differentiation effects of CIF stimulation and we strongly feel that it deserves an in-depth presentation and discussion. Additionally, there are striking resemblances of the CIF transcriptional profile with immune responses elicited by PAMP treatments, which we felt cannot be ignored and should be discussed.

Referee #3:

- general summary and opinion about the principle significance of the study, its questions and findings

These last years, the Geldner's lab, following a pioneering mutant screen, unravelled the Schengen mutants, affected in the deposition of the Casparian strip in plants. Through the characterization of these mutants and their corresponding proteins, emerged progressively a picture highlighting the critical importance of membrane micro-domains in the fine regulation of cell wall polymer deposition in plants, using the Casparian strip as a main model. This previous work pointed that SGN1, 3, and the CIF2 peptides may act in similar signalling pathways. Here, the authors provide a mechanistic evidence that SGN3 and SGN1 act in a similar signalling cascade, and that SGN3 is the most critical initiator of this pathway for precisely localized deposition of lignin in the Casparian strip through. The work elegantly reconciles the genetic and cellular evidence with kinase assays, highlighting the phosphorylation signalling hierarchy of SGN3 over downstream SGN1, leading to the subsequent activation of NOX proteins producing ROS, a necessary step for lignin polymerization. It allows a more comprehensive view of the Schengen pathway at signalling level, and will benefit the community when addressing the role of spatialization and signalling mechanisms at play during the critical steps of cell wall deposition during plant development.

However elegant is the study, we would like the authors to address the following points:

- specific major concerns essential to be addressed to support the conclusions

- minor concerns that should be addressed

-We wonder why the authors didn't assess whole RNA profiling after CIF treatments in *sgn1* background, as they are undertaking such analysis in the *sgn3* background and assessing the expression some specific genes in the *sgn1* background. This part of the manuscript is interesting to assess what genetic responses are triggered by the CIF signalling, and reaffirming that SGN3 is indeed the main receptor if CIF responses. However, assessing such responses in a more holistic manner in a *sgn1* background would allow to discriminate more precisely the genetic responses more specifically attributed to the SGN1 subsequent signalling pathway. Such experiment would be relevant to discuss this aspect.

REPLY: We agree that it would be interesting to also obtain transcriptional profiles of the *sgn1* mutant after CIF2 stimulation. We haven't done this because of constraints of budget, time and work capacity. More importantly, however, we would also predict that we would obtain a very similar gene set to the one we obtained by wild-type *sgn3* comparisons, simply less pronounced. In all the analyses we have done so far comparing *sgn1* and *sgn3*, we always observed the same, but weaker, intermediate defects or responses, compared to *sgn3*. We predict the same would be observed in a transcriptional profiling experiment. As pointed out in our reply to the next point, we postulate auxiliary homologue that can partially replace SGN1 in a *sgn1* mutant. We presently cannot exclude, but presently have no evidence supporting, a model whereby SGN1 would only mediate signaling in a discrete branch of the SGN3 pathway.

- The authors mentioned that probable SGN1 homologs can partially compensate for its absence in a *sgn1* background. Do they have any genetic evidence for this? (higher order mutant) If yes, similarly to the question raised above, RNA profiling in such background would be interesting to assess.

REPLY: One of the genetic evidences why we postulate SGN1 homologue is response of *sgn1* to the CIF peptide. As we presented in figure2, *sgn1* does respond to the CIF peptide but does not completely complement the gap of CASP domain, while the *cif1,2* double mutant was perfectly complemented by the peptide treatment, implying other factors are playing significant roles to transduce signal from the receptor. Furthermore, as presented in Doblaz et al. (2017), CIF2 also induced less lignification in *sgn1* than in WT, again suggesting SGN1 was indeed necessary to complete downstream responses, but that other factors yet to be identified could partially mediate this response. The fact that there are more than 40 homologs in the RLCKVII family (often also called PBLs) and the reports of redundancy among PBLs in immune signaling, makes it very likely that the remaining responses in *sgn1* are mediated by homologs within this family. Unfortunately, identification of such homolog(s) represents a project on its own.

- any additional non-essential suggestions for improving the study (which will be at the author's/editor's discretion)

- Can the authors explain the presence of possible ROS production in the *sgn3* background after CIF2 addition? (visible as black dots on the bottom left of Fig. 4D, *sgn3*; slightly statistically different when considering only *sgn3* background in Fig. 4E) Indeed, If SGN3 CIF interaction is necessary to trigger ROS production through NOX proteins, ROS shouldn't be detected.

REPLY: We agree with the reviewer that the picture chosen gives the impression of a slightly enhanced level of precipitates in *sgn3*. However, we really don't think this is significant (it is not statistically significant, as it falls in the same group as unstimulated wild-type and unstimulated *sgn3* in the ANOVA test) and we would not feel comfortable interpreting this in any way. The assay we are using does have a significant degree of variability, which is why we have gone to great length to develop a method of quantification and to present each data point on the graph. We hope that our presentation makes it evident that this method is very reliable, but only when strong differences in ROS production are observed, as is the case after CIF2 stimulation.

- A fluorescence quantification of the outer vs inner fluorescence of SGN1-Cit and myrpalm-SGN1-CIT would be interesting in order to appreciate in a more quantitative manner the broader localization of SGN1 after myrpalm addition (eg fluo ratio outer vs inner side in the 2 conditions).

REPLY: We considered this, but think that it is not necessary for the following reasons: Wild-type SGN1-Cit is very strictly localized, with no detectable signal at the inner endodermal side, which makes quantification non-pertinent (even difficult to interpret, because of very strong inner vs. outer ratio values that would vary a lot). Adding myr-palm to SGN1 leads to a weaker, but easily observable localization at the inner endodermal side, compared to the outer one. This could be quantified. However, the aim of this experiment was to obtain a qualitative, not quantitative, answer to the question: Does presence of SGN1 on the inner side lead to constitutive signaling as predicted by the barrier surveillance model. The answer is yes, since we observe appearance of inner-corner lignification that is never observed in wild-type at the same stage. This absence/presence difference of lignification upon SGN1 mislocalisation, renders the question of how strongly SGN1 has actually been mislocalised to be of secondary concern. Would this ectopic localization have been weak, variable or only quantitative, then we would have agreed with the reviewer that the level of "inner side" SGN1 would have been of interest to quantify (and enhance).

- A graphical abstract might be relevant to offer a more graphical overview of the main MS findings for non specialists.

REPLY: We fully agree and we have now attempted to do so in a new last Figure 7.

- A more detailed legend on the 2 panels of each kinase assays (Fig2. and Fig.5) would be appreciated to ease the reader understanding of these experiments.

REPLY: We admit that the legend is very short. We have now provided a more detailed one for both experiments.

- p4, 1136: We believe the authors made a typo and refer to Fig. 1C when it should refer to Fig. 1D.

REPLY: Thank you for pointing out the typo. We have corrected this.

2nd Editorial Decision

12th Feb 2020

Thank you for submitting a revised version of your manuscript. Unfortunately I still have not received comments from reviewer #3 on the revised manuscript. To my assessment, the issues indicated by the reviewers have been sufficiently addressed and there now remain only a few editorial issues that have to be finalised before I can extend formal acceptance of the manuscript.

2nd Revision - authors' response

21st Feb 2020

The authors performed the requested editorial changes.

3rd Editorial Decision

26th Feb 2020

Thank you for implementing the final edits in your manuscript. I am now pleased to inform you that your manuscript has been accepted for publication.

Corresponding Author Name: Satoshi Fujita and Niko Geldner

Journal Submitted to: the EMBO journal

Manuscript Number: 2019-103894